# ForkMerge: Mitigating Negative Transfer in Auxiliary-Task Learning

**Junguang Jiang**,[*] **Baixu Chen**,[*] **Junwei Pan**[§], **Ximei Wang**[§], **Dapeng Liu**[§], **Jie Jiang**[§],
**Mingsheng Long**[✉]
School of Software, BNRist, Tsinghua University, China
[§]Tencent Inc, China
{jjg20,cbx22}@mails.tsinghua.edu.cn, {jonaspan,messixmwang,rocliu,zeus}@tencent.com,
mingsheng@tsinghua.edu.cn

## Abstract

Auxiliary-Task Learning (ATL) aims to improve the performance of the target task by leveraging the knowledge obtained from related tasks. Occasionally, learning multiple tasks simultaneously results in lower accuracy than learning only the target task, which is known as negative transfer. This problem is often attributed to the gradient conflicts among tasks, and is frequently tackled by coordinating the task gradients in previous works. However, these optimization-based methods largely overlook the auxiliary-target generalization capability. To better understand the root cause of negative transfer, we experimentally investigate it from both optimization and generalization perspectives. Based on our findings, we introduce *ForkMerge*, a novel approach that periodically forks the model into multiple branches, automatically searches the varying task weights by minimizing target validation errors, and dynamically merges all branches to filter out detrimental task-parameter updates. On a series of auxiliary-task learning benchmarks, *ForkMerge* outperforms existing methods and effectively mitigates negative transfer.

## 1 Introduction

Deep neural networks have achieved remarkable success in various machine learning applications, such as computer vision [23, 22], natural language processing [62, 11, 57], and recommendation systems [46]. However, one major challenge in training deep neural networks is the scarcity of labeled data. In recent years, Auxiliary-Task Learning (ATL) has emerged as a promising technique to address this challenge [67, 39, 43]. ATL improves the generalization of target tasks by leveraging the useful signals provided by some related auxiliary tasks. For instance, larger-scale tasks, such as user click prediction, can be utilized as auxiliary tasks to improve the performance of smaller-scale target tasks, such as user conversion prediction in recommendation [47, 36]. Self-supervised tasks on unlabeled data can serve as auxiliary tasks to improve the performance of the target task in computer vision and natural language processing, without requiring additional labeled data [34, 69, 11, 3].

However, in practice, learning multiple tasks simultaneously sometimes leads to performance degradation compared to learning only the target task, a phenomenon known as *negative transfer* [84, 75]. Even in large language models, negative transfer problems may still exist. For example, RLHF [7], a key component of ChatGPT [57], achieves negative effects on nearly half of the multiple-choice question tasks when post-training GPT-4 [58]. There has been a significant amount of methods proposed to mitigate negative transfer in ATL [71, 79, 15, 39]. Notable previous studies attribute negative transfer to the optimization difficulty, especially the gradient conflicts between different tasks, and propose to mitigate negative transfer by reducing interference between task gradients

---

[*]Equal contribution.

37th Conference on Neural Information Processing Systems (NeurIPS 2023).

[79, 15]. Other works focus on selecting the most relevant auxiliary tasks and reducing negative transfer by avoiding task groups with severe task conflicts [71, 17]. However, despite the significant efforts to address negative transfer, its underlying causes are still not fully understood.

In this regard, we experimentally analyze potential causes of negative transfer in ATL from the perspectives of optimization and generalization. From an optimization view, our experiments suggest that *gradient conflicts do not necessarily lead to negative transfer*. For example, weight decay, a special auxiliary task, can conflict with the target task in gradients but still be beneficial to the target performance. From a generalization view, we observe that negative transfer is more likely to occur when *the distribution shift between the multi-task training data and target test data is enlarged.*

Based on our above findings, we present a new approach named *ForkMerge*. Since we cannot know which task distribution combination leads to better generalization in advance, and training models for each possible distribution is prohibitively expensive, we transform the problem of combining task distributions into that of combining model hypotheses. Specifically, we fork the model into multiple branches and optimize the parameters of different branches on diverse data distributions by varying the task weights. Then at regular intervals, we merge and synchronize the parameters of each branch to approach the optimal model hypothesis. In this way, we will filter out harmful parameter updates to mitigate negative transfer and keep desirable parameter updates to promote positive transfer.

The contributions of this work are summarized as follows: (1) We systematically identify the problem and analyze the causes of negative transfer in ATL. (2) We propose *ForkMerge*, a novel approach to mitigate negative transfer and boost the performance of ATL. (3) We conduct extensive experiments and validate that *ForkMerge* outperforms previous methods on a series of ATL benchmarks.

## 2 Related Work

### 2.1 Auxiliary-Task Learning

Auxiliary-Task Learning (ATL) enhances a model's performance on a target task by utilizing knowledge from related auxiliary tasks. The two main challenges in ATL are selecting appropriate auxiliary tasks and optimizing them jointly with the target task. To find the proper auxiliary tasks for ATL, recent studies have explored task relationships by grouping positively related tasks together and assigning unrelated tasks to different groups to avoid task interference [81, 71, 17, 70]. Once auxiliary tasks are determined, most ATL methods create a unified loss by linearly combining the target and auxiliary losses. However, choosing task weights is challenging due to the exponential increase in search space with the number of tasks, and fixing the weight of each task loss can lead to negative transfer [32]. Recent studies propose various methods to automatically choose task weights, such as using one-step or multi-step gradient similarity [15, 39, 9], minimizing representation-based task distance [2] or gradient gap [67], employing a parametric cascade auxiliary network [54], or from the perspective of bargaining game [66]. However, these methods mainly address the optimization difficulty after introducing auxiliary tasks and may overlook the generalization problem.

Recently, AANG [10] formulates a novel searching space of auxiliary tasks and adopts the meta-learning technique, which prioritizes target task generalization, to learn single-step task weightings. This parallel finding highlights the importance of the target task generalization and we further introduce the multi-step task weightings to reduce the estimation uncertainty. Another parallel method, ColD Fusion [12], explores collaborative multitask learning and proposes to fuse each contributor's parameter to construct a shared model. In this paper, we further take into account the diversity of tasks and the intricacies of task relationships and derive a method for combining model parameters from the weights of task combinations.

### 2.2 Multi-Task Learning

Different from ATL, Multi-Task Learning (MTL) aims to improve the performance of all tasks by learning multiple objectives from a shared representation. To facilitate information sharing and minimize task conflict, many multi-task architectures have been designed, including hard-parameter sharing [30, 22, 24] and soft-parameter sharing [51, 64, 16, 46, 44, 48, 72]. Another line of work aims to optimize strategies to reduce task conflict. Methods such as loss balancing and gradient balancing propose to find suitable task weighting through various criteria, such as task uncertainty [28], task loss magnitudes [44], gradient norm [5], and gradient directions [79, 6, 40, 41, 25, 55].

Although MTL methods can be directly used to jointly train auxiliary and target tasks, the asymmetric task relationships in ATL are usually not taken into account in MTL.

### 2.3 Negative Transfer

Negative Transfer (NT) is a widely existing phenomenon in machine learning, where transferring knowledge from the source data or model can have negative impact on the target learner [63, 60, 27]. To mitigate negative transfer, domain adaptation methods design importance sampling or instance weighting strategies to prioritize related source data [75, 83]. Fine-tuning methods filter out detrimental pre-trained knowledge by suppressing untransferable spectral components in the representation [4]. MTL methods use gradient surgery or task weighting to reduce the gradient conflicts across tasks [79, 76, 25, 42]. Different from previous work, we propose to dynamically filter out harmful parameter updates in the training process to mitigate negative transfer. Besides, we provide an in-depth experimental analysis of the causes of negative transfer in ATL, which is rare in this field yet will be helpful for future research.

## 3 Negative Transfer Analysis

**Problem and Notation.** In this section, we assume that both the target task $\mathcal{T}_{\text{tgt}}$ and the auxiliary task $\mathcal{T}_{\text{aux}}$ are given. Then the objective is to find model parameters $\theta$ that achieve higher performance on the target task by joint training with the auxiliary task,

$$\min_{\theta} \mathbb{E}_{\mathcal{T}_{\text{tgt}}} \mathcal{L}_{\text{tgt}}(\theta) + \lambda \mathbb{E}_{\mathcal{T}_{\text{aux}}} \mathcal{L}_{\text{aux}}(\theta), \tag{1}$$

where $\mathcal{L}$ is the training loss, and $\lambda$ is the relative weighting hyper-parameter between the auxiliary task and the target task. Our final objective is $\max_{\theta} [\mathcal{P}(\theta)]$, where $\mathcal{P}$ is the relative performance measure for the target task $\mathcal{T}_{\text{tgt}}$, such as the accuracy in classification. Next we define the Transfer Gain to measure the impact of $\mathcal{T}_{\text{aux}}$ on $\mathcal{T}_{\text{tgt}}$.

**Definition 3.1 (Transfer Gain, TG).** Denote the model obtained by some ATL algorithm $\mathcal{A}$ as $\theta_{\mathcal{A}}(\mathcal{T}_{\text{tgt}}, \mathcal{T}_{\text{aux}}, \lambda)$ and the model obtained by single-task learning on target task as $\theta(\mathcal{T}_{\text{tgt}})$. Let $\mathcal{P}$ be the performance measure on the target task $\mathcal{T}_{\text{tgt}}$. Then the algorithm $\mathcal{A}$ can be evaluated by

$$TG(\lambda, \mathcal{A}) = \mathcal{P}(\theta_{\mathcal{A}}(\mathcal{T}_{\text{tgt}}, \mathcal{T}_{\text{aux}}, \lambda)) - \mathcal{P}(\theta(\mathcal{T}_{\text{tgt}})). \tag{2}$$

Going beyond previous work on Negative Transfer (NT) [75, 84], we further divide negative transfer in ATL into two types.

**Definition 3.2 (Weak Negative Transfer, WNT).** For some ATL algorithm $\mathcal{A}$ with weighting hyper-parameter $\lambda$ , weak negative transfer occurs if $TG(\lambda, \mathcal{A}) < 0$.

**Definition 3.3 (Strong Negative Transfer, SNT).** For some ATL algorithm $\mathcal{A}$, strong negative transfer occurs if $\max_{\lambda > 0} TG(\lambda, \mathcal{A}) < 0$.

Figure 1 illustrates the difference between weak negative transfer and strong negative transfer. The most essential difference is that we might be able to avoid weak negative transfer by selecting a proper weighting hyper-parameter $\lambda$, yet we cannot avoid strong negative transfer in this way.

Next, we will analyze negative transfer in ATL from two different perspectives: **optimization** and **generalization**. We conduct our analysis on a multi-domain image recognition dataset DomainNet [61] with ResNet-18 [23] pre-trained on ImageNet. Specifically, we use task Painting and Quickdraw in DomainNet as target tasks respectively to showcase weak negative transfer and strong negative transfer, and mix all other tasks in DomainNet as auxiliary tasks. We will elaborate on the DomainNet dataset in Appendix C.3 and provide the detailed experiment design in Appendix B.

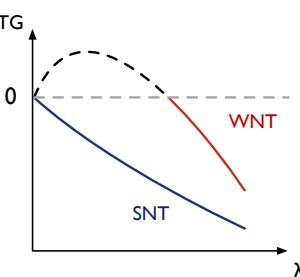

Figure 1: Weak Negative Transfer (WNT) vs. Strong Negative Transfer (SNT).

### 3.1 Effect of Gradient Conflicts

It is widely believed that gradient conflicts between different tasks will lead to optimization difficulties [79, 40], which in turn lead to negative transfer. The degree of gradient conflict is usually measured by the Gradient Cosine Similarity [79, 76, 15].

**Definition 3.4** (**Gradient Cosine Similarity, GCS**). Denote $\phi_{ij}$ as the angle between two task gradients $\mathbf{g}_i$ and $\mathbf{g}_j$, then we define the gradient cosine similarity as $\cos \phi_{ij}$ and the gradients as conflicting when $\cos \phi_{ij} < 0$.

In Figure 2, we plot the correlation curve between gradient cosine similarity and transfer gain. Somewhat counterintuitively, we observe that negative transfer and gradient conflicts are not strongly correlated, and negative transfer might be severer when the task gradients are highly consistent.

> *Finding 1. Negative transfer is not necessarily caused by gradient conflicts and gradient conflicts do not necessarily lead to negative transfer.*

It seems contradictory to the previous work [79, 15] and the reason is that previous work mainly considers the *optimization* convergence during training, while in our experiments we further consider the *generalization* during evaluation (transfer gain is estimated on the validation set). Although the conflicting gradient of the auxiliary task will increase the training loss of the target task and slow down its convergence speed [37], it may also play a role similar to regularization [32], reducing the over-fitting of the target task, thereby reducing its generalization error. To confirm our hypothesis, we repeat the above experiments with the auxiliary task replaced by $L_2$ regularization and observe a similar phenomenon as shown in Figure 2(c)-(d), which indicates that the gradient conflict in ATL is not necessarily harmful, as it may serve as a proper regularization.

Figure 2 also indicates that the weighting hyper-parameter $\lambda$ in ATL has a large impact on negative transfer. A proper $\lambda$ not only reduces negative transfer but also promotes positive transfer.

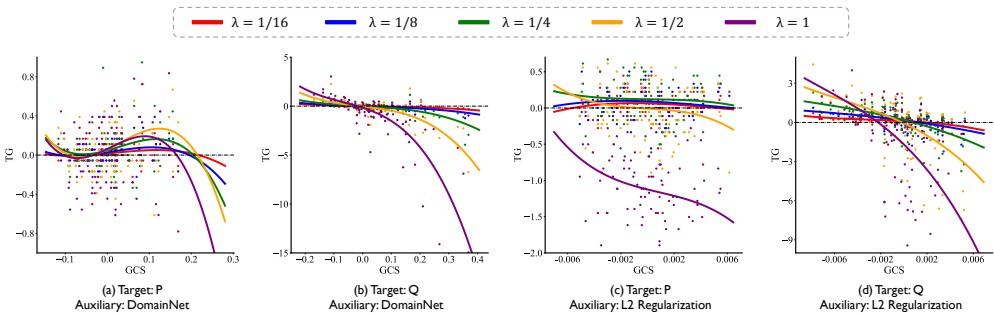

Figure 2: The effect of *gradient conflicts*. The correlation curve between Transfer Gain (TG) and Gradient Cosine Similarity (GCS) under different $\lambda$. For a fair comparison, each data point starts from the same model parameters in the middle of the training process and updates with one-step multi-task gradient descent. **P** and **Q** are short for **P**ainting and **Q**uickdraw tasks, respectively.

## 3.2 Effect of Distribution Shift

Next, we will analyze negative transfer from the perspective of generalization. We notice that adjusting $\lambda$ will change the data distribution that the model is fitting. For instance, when $\lambda = 0$, the model only fits the data distribution of the target task, and when $\lambda = 1$, the model will fit the interpolated distribution of the target and auxiliary tasks. Formally, given the target distribution $\mathcal{T}_{\text{tgt}}$ and the auxiliary distribution $\mathcal{T}_{\text{aux}}$, the interpolated distribution of the target and auxiliary task is $\mathcal{T}_{\text{inter}}$,

$$\mathcal{T}_{\text{inter}} \sim (1 - Z)\mathcal{T}_{\text{tgt}} + Z\mathcal{T}_{\text{aux}}, Z \sim \text{Bernoulli}(\frac{\lambda}{1 + \lambda}), \tag{3}$$

where $\lambda$ is the task-weighting hyper-parameter. Figure 3(a) quantitatively visualizes the distribution shift under different $\lambda$ using t-SNE [74].

To quantitatively measure the distribution shift in ATL, we introduce the following definitions. Following the notations of [53], we consider multiclass classification with hypothesis space $\mathcal{F}$ of scoring functions $f : \mathcal{X} \times \mathcal{Y} \rightarrow \mathbb{R}$ where $f(x, y)$ indicates the confidence of predicting $x$ as $y$.

**Definition 3.5** (**Confidence Score Discrepancy, CSD**). Given scoring function hypothesis $\mathcal{F}$, denote the optimal hypothesis on distribution $\mathcal{D}$ as $f_{\mathcal{D}}^*$, then confidence score discrepancy between

distribution $\mathcal{D}$ and $\mathcal{D}'$ induced by $\mathcal{F}$ is defined by

$$d_{\mathcal{F}}(\mathcal{D}, \mathcal{D}') \triangleq 1 - \mathbb{E}_{x \sim \mathcal{D}'} \max_{y \in \mathcal{Y}} f_{\mathcal{D}}^*(x, y). \tag{4}$$

Confidence score discrepancy between training and test data indicates how unconfident the model is on the test data, which is expected to increase when the data shift enlarges [59, 50].

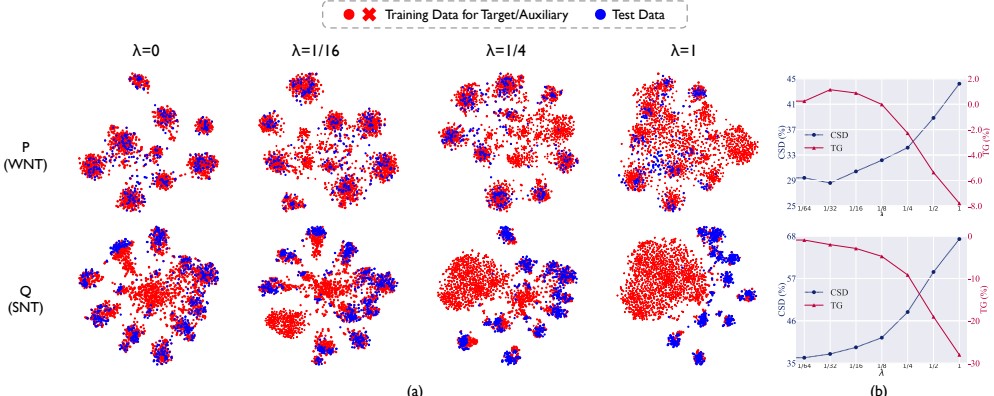

Figure 3: The effect of *distribution shift*. **(a)** Visualization of training distribution and test distribution under different $\lambda$. **(b)** For weak negative transfer tasks, as $\lambda$ increases, Confidence Score Discrepancy (CSD) first drops and then rises and Transfer Gain (TG) is first positive and then negative. For strong negative transfer tasks, CSD increases monotonically and TG remains negative.

Figure 3(b) indicates the correlation between confidence score discrepancy and transfer gain. For weak negative transfer tasks, when $\lambda$ increases at first, the introduced auxiliary tasks will shift the training distribution towards the test distribution, thus decreasing the confidence score discrepancy between training and test data and improving the generalization of the target task. However, when $\lambda$ continues to increase, the distribution shift gradually increases, finally resulting in negative transfer. For strong negative transfer tasks, there is a large gap between the distribution of the introduced auxiliary tasks and that of the target task. Thus, increasing $\lambda$ always enlarges confidence score discrepancy and always leads to negative transfer. In summary,

> *Finding 2. Negative transfer is likely to occur if the introduced auxiliary task enlarges the distribution shift between training and test data for the target task.*

## 4 Methods

In Section 4.1, based on our above analysis, we will introduce how to mitigate negative transfer when the auxiliary task is determined. Then in Section 4.2, we will further discuss how to use the proposed method to select appropriate auxiliary tasks and optimize them jointly with the target task simultaneously.

### 4.1 ForkMerge

In this section, we assume that the auxiliary task $\mathcal{T}_{aux}$ is given. When updating the parameters $\theta_t$ with Equation (1) at training step $t$, we have

$$\theta_{t+1}(\lambda) = \theta_t - \eta(\mathbf{g}_{tgt}(\theta_t) + \lambda \mathbf{g}_{aux}(\theta_t)), \tag{5}$$

where $\eta$ is the learning rate, $\mathbf{g}_{tgt}$ and $\mathbf{g}_{aux}$ are the gradients calculated from $\mathcal{L}_{tgt}$ and $\mathcal{L}_{aux}$ respectively. Section 3.1 reveals that the gradient conflict between $\mathbf{g}_{tgt}$ and $\mathbf{g}_{aux}$ does not necessarily lead to negative transfer as long as $\lambda$ is carefully tuned and Section 3.2 shows that negative transfer is related to generalization. Thus we propose to dynamically adjust $\lambda$ according to the target validation performance $\widehat{\mathcal{P}}$ to mitigate negative transfer:

$$\max_{\lambda} \widehat{\mathcal{P}}(\theta_{t+1}) = \widehat{\mathcal{P}}(\theta_t - \eta(\mathbf{g}_{tgt}(\theta_t) + \lambda \mathbf{g}_{aux}(\theta_t))). \tag{6}$$

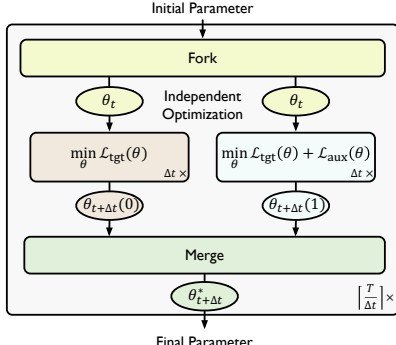

Initial Parameter

Final Parameter

Figure 4: ForkMerge training pipeline. The model parameters will be forked into two branches, one optimized with the target task loss and the other jointly trained, and be merged at regular intervals of $\Delta t$ steps.

**Algorithm 1** ForkMerge Training Pipeline.

**Require:** initial model parameter $\theta_0$, total iterations $T$, interval $\Delta t$
**Ensure:** final model parameter $\theta_T^*$, task relevance $\lambda^*$
    fork model into 2 copies $\{\theta^b\}_{b=0}^1$
    **for** $b = 0$ **to** 1 **do**
        $\theta_0^b \leftarrow \theta_0$          ▷ initialization
    **end for**
    **while** $t < T$ **do**
        **for** $b = 0$ **to** 1 **do**          ▷ independent update
            **for** $t' = t$ **to** $t + \Delta t - 1$ **do**
                $\theta_{t'+1}^b = \theta_{t'}^b - \eta(\mathbf{g}_{\text{tgt}}(\theta_{t'}^b) + b \cdot \mathbf{g}_{\text{aux}}(\theta_{t'}^b))$
            **end for**
        **end for**
        $\lambda^* \leftarrow \arg\max_\lambda \widehat{\mathcal{P}}((1 - \lambda)\theta_{t+\Delta t}^0 + \lambda\theta_{t+\Delta t}^1)$
                         ▷ search $\lambda$ on the validation set
        $\theta_{t+\Delta t}^* \leftarrow (1 - \lambda^*)\theta_{t+\Delta t}^0 + \lambda^*\theta_{t+\Delta t}^1$
                         ▷ merge parameters
        **for** $b = 0$ **to** 1 **do**
            $\theta_{t+\Delta t}^b \leftarrow \theta_{t+\Delta t}^*$        ▷ synchronize parameters
        **end for**
        $t \leftarrow t + \Delta t$
    **end while**

Equation (6) is a bi-level optimization problem. One common approach is to first approximate $\widehat{\mathcal{P}}$ with the loss of a batch of data on the validation set, and then use first-order approximation to solve $\lambda$ [18, 43]. However, these approximations within a single step of gradient descent introduce large noise to the estimation of $\lambda$ and also increase the risk of over-fitting the validation set. To tackle these issues, we first rewrite Equation (6) equally as

$$\max_\lambda \widehat{\mathcal{P}}\big((1 - \lambda)\theta_{t+1}(0) + \lambda\theta_{t+1}(1)\big), \tag{7}$$

where $\theta_{t+1}(0) = \theta_t - \eta\mathbf{g}_{\text{tgt}}(\theta_t)$ and $\theta_{t+1}(1) = \theta_t - \eta(\mathbf{g}_{\text{tgt}}(\theta_t) + \mathbf{g}_{\text{aux}}(\theta_t))$. The proof is in Appendix A.1. Note that we assume the optimal $\lambda^*$ satisfies $0 \leq \lambda^* \leq 1$, which can be guaranteed by increasing the scale of $\mathcal{L}_{\text{aux}}$ when necessary. Yet an accurate estimation of performance $\widehat{\mathcal{P}}$ in Equation (7) is still computationally expensive and prone to over-fitting, thus we extend the one gradient step to $\Delta t$ steps,

$$\lambda^* = \arg\max_\lambda \widehat{\mathcal{P}}\big((1 - \lambda)\theta_{t+\Delta t}(0) + \lambda\theta_{t+\Delta t}(1)\big). \tag{8}$$

**Algorithm.** As shown in Figure 4 and Algorithm 1, the initial model parameters $\theta_t$ at training step $t$ will first be forked into two branches. The first one will be optimized only with the target task loss $\mathcal{L}_{\text{tgt}}$ for $\Delta t$ iterations to obtain $\theta_{t+\Delta t}(0)$, while the other one will be jointly trained for $\Delta t$ iterations to obtain $\theta_{t+\Delta t}(1)$. Then we will search the optimal $\lambda^*$ that linearly combines the above two sets of parameters to maximize the validation performance $\widehat{\mathcal{P}}$. When weak negative transfer occurs in the joint training branch, we can select a proper $\lambda^*$ between $0$ and $1$. And when strong negative transfer occurs, we can simply set $\lambda^*$ to $0$. Finally, the newly merged parameter $\theta_{t+\Delta t}^* = (1 - \lambda^*)\theta_{t+\Delta t}(0) + \lambda^*\theta_{t+\Delta t}(1)$ will join in a new round, being forked into two branches again and repeating the optimization process for $\lceil \frac{T}{\Delta t} \rceil$ times.

**Discussion.** Compared to grid searching $\lambda$, which is widely used in practice, ForkMerge can dynamically transfer knowledge from auxiliary tasks to the target task during training with varying $\lambda^*$. In terms of computation cost, ForkMerge has a lower complexity as it only requires training 2 branches while grid searching has a cost proportional to the number of hyper-parameters to be searched.

### 4.2 ForkMerge for Task Selection Simultaneously

When multiple auxiliary tasks are available, we can simply mix all the auxiliary tasks together to form a single auxiliary task. This simple strategy actually works well in most scenarios (see Section 5.2) and is computationally cheap. However, when further increasing the performance is desired, we can also dynamically select the optimal weighting for each auxiliary task. Formally, the objective

when optimizing the model for the target task $\mathcal{T}_0$ with multiple auxiliary tasks $\{\mathcal{T}_k\}_{k=1}^K$ is

$$\min_\theta \mathbb{E}_{\mathcal{T}_0}\mathcal{L}_0(\theta) + \sum_{k=1}^K \lambda_k \mathbb{E}_{\mathcal{T}_k}\mathcal{L}_k(\theta), \tag{9}$$

where $\sum_{k=1}^K \lambda_k \leq 1$ and $\forall k, \lambda_k \geq 0$. Using gradient descent to update $\theta_t$ at training step $t$, we have

$$\theta_{t+1}(\boldsymbol{\lambda}) = \theta_t - \eta \sum_{k=0}^K \lambda_k \mathbf{g}_k(\theta_t), \tag{10}$$

where $\lambda_0 = 1$. Given $K$ task-weighting vectors $\{\boldsymbol{\omega}^k\}_{k=0}^K$ that satisfies $\omega_i^k = \mathbb{1}[i = k \text{ or } i = 0]$, i.e., the $k$-th and 0-th dimensions of $\boldsymbol{\omega}^k$ are 1 and the rest are 0, and a vector $\boldsymbol{\Lambda}$ that satisfies

$$\Lambda_k = \begin{cases} 1 - \sum_{i \neq 0} \lambda_i, & k = 0, \\ \\ \lambda_k, & k \neq 0, \end{cases} \tag{11}$$

then optimizing $\boldsymbol{\lambda}^*$ in Equation (10) is equivalent to

$$\boldsymbol{\Lambda}^* = \arg\max_{\boldsymbol{\Lambda}} \widehat{\mathcal{P}}\Big(\sum_{k=0}^K \Lambda_k \theta_{t+1}(\boldsymbol{\omega}^k)\Big). \tag{12}$$

In Equation (12), the initial model parameters are forked into $K + 1$ branches, where one branch is optimized with the target task, and the other branches are jointly optimized with one auxiliary task and the target task. Then we find the optimal $\boldsymbol{\Lambda}^*$ that linearly combines the $K + 1$ sets of parameters to maximize the validation performance (see proof of Equation (12) and the detailed algorithm in Appendix A.2). The training computational complexity of Equation 12 is $\mathcal{O}(K)$, which is much lower than the exponential complexity of grid searching, but still quite large. Inspired by the early-stop approximation used in task grouping methods [71], we can prune the forking branches with $\Lambda_k = 0$ (strong negative transfer) and only keep the branches with the largest $K' < K$ values in $\boldsymbol{\Lambda}$ after the early merge step. In this way, those useless branches with irrelevant auxiliary tasks can be stopped early. Additionally, we introduce a greedy search strategy in Algorithm 3 to further reduce the computation complexity when grid searching all possible values of $\boldsymbol{\Lambda}$.

Lastly, we introduce a general form of ForkMerge. Assuming $B$ candidate branches with task-weighting vectors $\boldsymbol{\nu}^b$ ($b = 1, \dots, B$), the goal is to optimize $\overline{\boldsymbol{\Lambda}}^*$:

$$\overline{\boldsymbol{\Lambda}}^* = \arg\max_{\overline{\boldsymbol{\Lambda}}} \widehat{\mathcal{P}}\Big(\sum_{b=1}^B \overline{\Lambda}_b \theta_{t+\Delta t}(\boldsymbol{\nu}^b)\Big). \tag{13}$$

From a generalization view, the mixture distributions constructed by different $\boldsymbol{\nu}$ lead to diverse data shifts from the target distribution, yet we cannot predict which $\boldsymbol{\nu}$ leads to better generalization. Thus, we transform the problem of mixture distribution into that of mixture hypothesis [49] and the models trained on different distributions are combined dynamically via $\overline{\boldsymbol{\Lambda}}^*$ to approach the optimal parameters. Here, Equation 12 is a particular case by substituting $B = K + 1$ and $\nu_i^b = \mathbb{1}[i = b - 1 \text{ or } i = 0]$. By comparison, Equation 13 allows us to introduce human prior into ForkMerge by constructing more efficient branches, and also provides possibilities for combining ForkMerge with previous task grouping methods [81, 71, 17]. The detailed algorithm of Equation 13 can be found in Algorithm 2.

## 5 Experiments

We evaluate the effectiveness of ForkMerge under various settings, including multi-task learning, multi-domain learning, and semi-supervised learning. First, in Section 5.1, we illustrate the prevalence of negative transfer and explain how ForkMerge can mitigate this problem. In Section 5.2, We examine whether ForkMerge can mitigate negative transfer when joint training the auxiliary and target tasks, and compare it with other methods. In Section 5.3, we further use ForkMerge for task selection simultaneously. Experiment details can be found in Appendix C. We will provide additional analysis and comparison experiments in Appendix D. The codebase for both our method and the compared methods will be available at `https://github.com/thuml/ForkMerge`.

## 5.1 Motivation Experiment

**Negative Transfer is widespread across different tasks.** In Figure 5 (a), we visualize the transfer gains between 30 task pairs on DomainNet, where the auxiliary and target tasks are equally weighted, and we observe that negative transfer is common in such case (23 of 30 combinations lead to negative transfer). Besides, as mentioned in Definition 3.2 and 3.3, whether negative transfer occurs is related to a specific ATL algorithm, in Figure 5 (b), we observe that negative transfer in all 30 combinations can be successfully avoided when we use ForkMerge algorithm. This observation further indicates the limitation of task grouping methods [71, 17], since they use Equal Weight between tasks and may discard some useful auxiliary tasks.

**Mixture of hypotheses is an approximation of mixture of distribution.** Figure 6 uses the ternary heatmaps to visualize the linear combination of a set of three models optimized with different task weightings for $25K$ iterations, including a single-task model and two multi-task models. Similar to mixing distributions for weak negative transfer task Painting (see Figure 3), the transfer gain when mixing models **P**ainting and **P**ainting+**R**eal first increases and then decreases. Also similar to mixing distributions for strong negative transfer task **Q**uickdraw, the transfer gain when mixing models **Q**uickdraw and **Q**uickdraw+**R**eal decreases monotonically. Besides, Figure 6 also indicates a good property of deep models: the loss surfaces of over-parameterized deep neural networks are quite well-behaved and smooth after convergence, which has also been mentioned by previous works [20, 35] and provides an intuitive explanation of the merge step in ForkMerge.

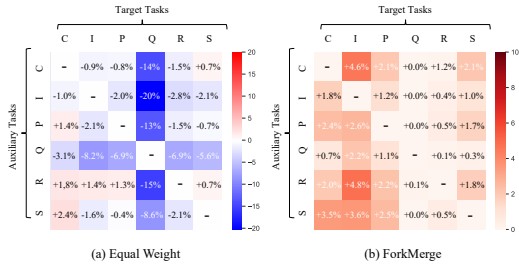

(a) Equal Weight          (b) ForkMerge

Figure 5: Negative Transfer on DomainNet. The rows of each matrix represent auxiliary tasks, and the columns represent target tasks. The blue and red cells correspond to negative and positive transfer gain. Deeper colors indicate stronger impacts.

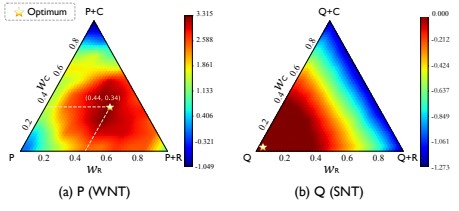

(a) P (WNT)          (b) Q (SNT)

Figure 6: Ternary heatmap for mixture of model hypotheses. Each triangle vertex represents an optimized model, e.g., **P+R** is the model jointly optimized with **P**ainting and **R**eal tasks. Each point inside the triangle corresponds to a mixture of model hypotheses and its heat value measures the Transfer Gain (TG).

## 5.2 Use ForkMerge for Joint Optimization

First, we use ForkMerge only for joint training of the target and auxiliary tasks. When datasets contain multiple tasks, we will mix all tasks together to form a single auxiliary task for ForkMerge. Yet for the compared methods, a distinction is still made between different tasks for better performance.

Specifically, we compare ForkMerge with: (1) Single Task Learning (STL); (2) EW, which assigns equal weight to all tasks; (3) GCS [15], an ATL approach using gradient similarity between target and auxiliary tasks; (4) OL_AUX [39], an ATL approach adjusting the loss weight based on gradient inner product; (5) ARML [67], an ATL approach adjusting the loss weight based on gradient difference; (6) Auto-$\lambda$ [43], an ATL method that estimates loss weight through finite-difference approximation [18]; (7) Post-train, an ATL method that pre-trains the model on all tasks and then fine-tunes it for each task separately. (8) UW [28], which adjusts weights based on task uncertainty; (9) DWA [44], which adjusts weights based on loss change; (10) MGDA [65], which computes a convex combination of gradients with a minimum norm to balance tasks; (11) GradNorm [5], which rescales the gradient norms of different tasks to the same range; (12) PCGrad [79], which eliminates conflicting gradient components; (13) IMTL [41], which uses an update direction with equal projections on task gradients; (14) CAGrad [40], which optimizes for the average loss and minimum decrease rate across tasks; (15) NashMTL [55], which combines the gradients using the Nash bargaining solution. Since different tasks have varying evaluation metrics, we will report the average per-task performance improvement for each method using $\Delta_m$, as defined in Appendix C.1.

**Auxiliary-Task Scene Understanding.** We evaluate on the widely-used multi-task scene understanding dataset, NYUv2 [68], which contains 3 tasks: 13-class semantic segmentation, depth estimation, and surface normal prediction. Following [55], we use 636, 159 and 654 images for training, validation, and test. Our implementation is based on LibMTL [38] and MTAN [44]. The results are presented in Table 1. Negative transfer is not severe on this dataset, where both *segmentation* and *depth* benefit from ATL and only *normal* task gets worse. In such cases, our method still achieves significant improvement on all tasks. We also find that Post-train serves as a strong baseline in most of our ATL experiments. Its drawback is that it fails to consider the task relationship in the pre-training phase, and suffers from catastrophic forgetting during the fine-tuning process.

**Auxiliary-Domain Image Recognition.** Further, we evaluate on the widely-used multi-domain image recognition dataset, DomainNet [61], which contains 6 diverse visual domains and approximately 0.6 million images distributed among 345 categories, where the task difference is reflected in the marginal distribution. Our implementation is based on TLlib [26]. As the original DomainNet does not provide a separate validation set, we randomly split 50% data from the test set as the validation set. The results are presented in Table 2. DomainNet contains both positive transfer tasks (**C**lipart), weak negative transfer tasks (**I**nfograph, **P**ainting, **R**eal, **S**ketch), and strong negative transfer tasks (**Q**uickdraw). When negative transfer occurs, previous ATL methods lead to severe performance degradation, while our method can automatically avoid strong negative transfer and improve the performance over STL in other cases.

Table 1: Performance on NYUv2 dataset.

| Methods | Segmentation | | Depth | | Normal | $\Delta_m\uparrow$ |
|---|---|---|---|---|---|---|
| | mIoU↑ | Pix Acc↑ | Abs Err↓ | Rel Err↓ | Mean↓ | |
| STL | 51.42 | 74.14 | 41.74 | 17.37 | 22.82 | - |
| EW | 52.13 | 74.51 | 39.03 | 16.43 | 24.14 | 0.30% |
| UW | 52.51 | 74.72 | 39.15 | 16.56 | 23.99 | 0.63% |
| DWA | 52.10 | 74.45 | 39.26 | 16.57 | 24.12 | 0.07% |
| MGDA | 50.79 | 73.81 | 39.19 | 16.25 | 23.14 | 1.44% |
| GradNorm | 52.25 | 74.54 | 39.31 | 16.37 | 23.86 | 0.56% |
| PCGrad | 51.77 | 74.72 | **38.91** | 16.36 | 24.31 | 0.22% |
| IMTL | 52.24 | 74.73 | 39.46 | **15.92** | 23.25 | 2.10% |
| CAGrad | 52.04 | 74.25 | 39.06 | 16.30 | 23.39 | 1.41% |
| NashMTL | 51.73 | 74.10 | 39.55 | 16.50 | 23.21 | 1.11% |
| GCS | 52.67 | 74.59 | 39.72 | 16.64 | 24.10 | 0.09% |
| OL_AUX | 52.07 | 74.28 | 39.32 | 16.30 | 23.98 | 0.17% |
| ARML | 52.73 | 74.85 | 39.61 | 16.65 | 23.89 | 0.37% |
| Auto-$\lambda$ | 52.40 | 74.62 | 39.25 | 16.25 | 23.38 | 1.17% |
| Post-train | 52.08 | 74.86 | 39.58 | 16.77 | 22.98 | 1.49% |
| ForkMerge | **53.67** | **75.64** | 38.91 | 16.47 | **22.18** | **4.03%** |

Table 2: Performance on DomainNet dataset.

| Methods | C | I | P | Q | R | S | Avg | $\Delta_m\uparrow$ |
|---|---|---|---|---|---|---|---|---|
| STL | 77.6 | 41.4 | 71.8 | **73.0** | 84.6 | 70.2 | 69.8 | - |
| EW | 78.0 | 38.1 | 67.2 | 50.8 | 77.1 | 67.0 | 63.0 | -9.62% |
| UW | 79.1 | 35.8 | 68.2 | 50.5 | 77.9 | 67.0 | 63.1 | -9.98% |
| DWA | 78.3 | 38.2 | 67.8 | 51.4 | 77.3 | 67.2 | 63.4 | -9.15% |
| MGDA | 78.1 | 37.2 | 69.2 | 51.0 | 80.0 | 67.3 | 63.8 | -8.80% |
| GradNorm | 78.4 | 38.9 | 69.4 | 52.9 | 79.0 | 67.7 | 64.4 | -7.68% |
| PCGrad | 78.3 | 38.0 | 68.2 | 50.4 | 77.4 | 67.3 | 63.3 | -9.32% |
| IMTL | 79.4 | 38.6 | 68.6 | 53.7 | 79.3 | 67.6 | 64.5 | -7.55% |
| CAGrad | 79.1 | 38.6 | 69.4 | 53.6 | 79.8 | 67.6 | 64.7 | -7.35% |
| NashMTL | 71.8 | 32.9 | 62.2 | 39.5 | 73.5 | 61.4 | 56.9 | -18.8% |
| GCS | 74.6 | 36.0 | 67.6 | 56.6 | 76.4 | 62.3 | 62.3 | -11.0% |
| OL_AUX | 68.2 | 33.5 | 65.3 | 54.1 | 76.3 | 60.9 | 59.7 | -14.8% |
| ARML | 75.6 | 36.8 | 67.8 | 52.4 | 77.6 | 64.2 | 62.4 | -10.7% |
| Auto-$\lambda$ | 78.3 | 37.8 | 70.2 | 56.3 | 79.7 | 67.1 | 64.9 | -7.18% |
| Post-train | 78.7 | 42.3 | 72.7 | **73.0** | 84.7 | 71.2 | 70.4 | +1.07% |
| ForkMerge | **79.9** | **42.7** | **73.5** | **73.0** | **85.2** | **72.0** | **71.1** | **+2.00%** |

### 5.3 Use ForkMerge for Task Selection Simultaneously

As mentioned in Section 4.2, when there are multiple auxiliary task candidates, we can use ForkMerge to simultaneously select auxiliary tasks and jointly train them with the target task, which is denoted as ForkMerge[‡].

**Auxiliary-Task Scene Understanding.** In NYUv2, we have 2 auxiliary tasks for any target task, thus we can construct 3 branches with different task weights in Equation 12. In this way, we are able to select auxiliary tasks adaptively by learning different $\Lambda$ for different branches in the merge step. As shown in Table 3, this strategy yields better overall performance.

**Auxiliary-Domain Image Recognition.** For any target task in DomainNet, we can construct up to 6 branches with different task weights in Equation 12, which is computationally expensive. As mentioned in Section 4.2, we will prune the branches after the first merge step to reduce the computation cost. Table 4 reveals the impact of the pruning strategy. As the number of branches increases, the gain brought by auxiliary tasks will enlarge, while the gain brought by each branch will reduce. Therefore, pruning is an effective strategy to achieve a better balance between performance and efficiency. In practical terms, when confronted with multiple auxiliary tasks, users have the flexibility to tailor the number of branches to align with their available computational resources.

**CTR and CTCVR Prediction.** We evaluate on AliExpress dataset [36], a recommendation dataset from the industry, which consists of 2 tasks: CTR and CTCVR, and 4 scenarios and more than 100M records. Our implementation is based on MTReclib [85]. For any target task in AliExpress, we can construct up to 8 branches with different task weights, and we prune to 3 branches after the first merge

Table 3: Effect of of branch number on NYUv2. Table 4: Effect of branch number on DomainNet.

| Methods | $B$ | Segmentation | | Depth | | Normal | $\Delta_m\uparrow$ |
|---|---|---|---|---|---|---|---|
| | | mIoU↑ | Pix Acc↑ | Abs Err↓ | Rel Err↓ | Mean↓ | |
| STL | 1 | 51.42 | 74.14 | 41.74 | 17.37 | 22.82 | - |
| EW | - | 52.13 | 74.51 | 39.03 | 16.43 | 24.14 | 0.30% |
| ForkMerge | 2 | 53.67 | 75.64 | 38.91 | 16.47 | **22.18** | 4.03% |
| ForkMerge[‡] | 3 | **54.30** | **75.78** | **38.42** | **16.11** | 22.41 | **4.59%** |

| Methods | $B$ | C | I | P | Q | R | S | $\Delta_m\uparrow$ | $\frac{\Delta_m}{B-1}\uparrow$ |
|---|---|---|---|---|---|---|---|---|---|
| STL | 1 | 77.6 | 41.4 | 71.8 | 73.0 | 84.6 | 70.2 | - | - |
| ForkMerge | 2 | 79.9 | 42.7 | 73.5 | 73.0 | 85.2 | 72.0 | 2.0% | 2.0% |
| | 3 | 81.1 | 44.0 | 73.7 | 73.0 | 85.2 | 72.7 | 3.0% | 1.5% |
| ForkMerge[‡] | 4 | 81.1 | 44.2 | 74.4 | 73.1 | 85.3 | 73.0 | 3.3% | 1.1% |
| | 6 | **81.3** | **44.4** | **74.7** | **73.2** | **85.3** | **73.4** | **3.6%** | 0.7% |

step. The results are presented in Table 5. Note that improving on such a large-scale dataset with auxiliary tasks is quite difficult. Still, ForkMerge achieves the best performance with $\Delta_m = \mathbf{1.30\%}$.

**Semi-Supervised Learning (SSL).** We also evaluate on two SSL datasets, CIFAR-10 [31] and SVHN [56]. Following [67], we use Self-supervised Semi-supervised Learning (S4L) [82] as our baseline algorithm and use 2 self-supervised tasks, Rotation [19] and Exempler-MT [14], as our auxiliary tasks. Table 6 presents the test error of S4L using different ATL approaches, along with other SSL methods, and shows that ForkMerge consistently outperforms the compared ATL methods. Note that we do not aim to propose a novel or state-of-the-art SSL method in this paper. Instead, we find that some SSL methods use ATL and the auxiliary task weights have a great impact (see Grid Search in Table 6). Thus, we use ForkMerge to improve the auxiliary task training within the context of SSL.

Table 5: Performance on AliExpress dataset.

| Methods | CTR | | | | CTCVR | | | | Avg | $\Delta_m\uparrow$ |
|---|---|---|---|---|---|---|---|---|---|---|
| | ES | FR | NL | US | ES | FR | NL | US | | |
| STL | 0.7299 | 0.7316 | 0.7237 | 0.7077 | 0.8778 | 0.8682 | 0.8652 | 0.8659 | 0.7963 | - |
| EW | 0.7299 | 0.7300 | 0.7248 | 0.7008 | 0.8855 | 0.8516 | 0.8606 | 0.8618 | 0.7931 | -0.39% |
| UW | 0.7276 | 0.7235 | 0.7250 | 0.7048 | 0.8814 | 0.8709 | 0.8599 | **0.8793** | 0.7966 | +0.00% |
| DWA | 0.7317 | 0.7284 | 0.7297 | 0.7061 | 0.8663 | 0.8695 | 0.8696 | 0.8484 | 0.7937 | -0.28% |
| MGDA | 0.6985 | 0.6926 | 0.7000 | 0.6676 | 0.8215 | 0.8145 | 0.7978 | 0.7917 | 0.7480 | -5.94% |
| GradNorm | 0.7239 | 0.7178 | 0.7101 | 0.7035 | 0.8851 | 0.8671 | 0.8465 | 0.8685 | 0.7903 | -0.79% |
| PCGrad | 0.7209 | 0.7193 | 0.7199 | 0.6892 | 0.8563 | 0.8621 | 0.8479 | 0.8413 | 0.7821 | -1.76% |
| IMTL | 0.7203 | 0.7193 | 0.7268 | 0.6852 | 0.8472 | 0.8502 | 0.8481 | 0.8282 | 0.7782 | -2.20% |
| CAGrad | 0.7280 | 0.7271 | 0.7223 | 0.6996 | 0.8712 | 0.8650 | 0.8417 | 0.8648 | 0.7900 | -0.77% |
| NashMTL | 0.7229 | 0.7245 | 0.7272 | 0.6972 | 0.8562 | 0.8606 | 0.8667 | 0.8497 | 0.7881 | -1.00% |
| GCS | 0.7229 | 0.7245 | 0.7272 | 0.6972 | 0.8562 | 0.8606 | 0.8667 | 0.8497 | 0.7881 | -0.49% |
| OL_AUX | 0.7311 | 0.7211 | 0.7239 | 0.7050 | 0.8779 | 0.8651 | 0.8610 | 0.8727 | 0.7947 | +0.54% |
| ARML | 0.7278 | 0.7247 | 0.7236 | 0.7030 | 0.8780 | 0.8671 | 0.8678 | 0.8670 | 0.7949 | +0.55% |
| Auto-$\lambda$ | 0.7282 | 0.7282 | 0.7263 | **0.7114** | 0.8852 | 0.8646 | 0.8640 | 0.8750 | 0.7979 | +0.19% |
| Post-train | 0.7291 | 0.7227 | 0.7244 | 0.7086 | 0.8889 | **0.8808** | 0.8654 | 0.8613 | 0.7977 | +0.14% |
| ForkMerge[‡] | **0.7402** | **0.7427** | **0.7416** | 0.7069 | **0.8928** | 0.8786 | **0.8753** | 0.8752 | **0.8067** | **+1.30%** |

Table 6: Peformance (test error) on CIFAR-10 and SVHN datasets.

| Methods | CIFAR-10 (4000 labels) | SVHN (1000 labels) | $\Delta_m\uparrow$ |
|---|---|---|---|
| STL | 20.3 | 12.80 | - |
| Π-Model [33] | 16.4 | 7.19 | 31.5% |
| Mean Teacher [73] | 15.9 | 5.65 | 38.8% |
| VAT [52] | 13.9 | 5.63 | 43.8% |
| Pseudo-Label [34] | 17.8 | 7.62 | 26.4% |
| S4L + EW | 15.7 | 7.83 | 30.7% |
| S4L + GradNorm | 14.1 | 7.68 | 35.3% |
| S4L + GCS | 15.0 | 7.02 | 35.6% |
| S4L + OL_AUX | 16.1 | 7.82 | 29.8% |
| S4L + ARML | 13.7 | 5.89 | 43.2% |
| S4L + Auto-$\lambda$ | 14.2 | 6.14 | 41.0% |
| S4L + Post-train | 15.8 | 7.85 | 30.4% |
| S4L + Grid Search | 13.8 | 6.07 | 42.3% |
| S4L + ForkMerge[‡] | **13.1** | **5.49** | **46.3%** |

## 6   Conclusion

Methods have been proposed to mitigate negative transfer in auxiliary-task learning, yet there still lacks an in-depth experimental analysis on the causes of negative transfer. In this paper, we systematically delved into the negative transfer issues and presented ForkMerge, an approach to enable auxiliary-task learning with positive transfer gains. Experimentally, ForkMerge achieves state-of-the-art accuracy on four different auxiliary-task learning benchmarks, while being computationally efficient. We view the integration of previous task grouping methods with our auxiliary task learning approach as a promising avenue for further research.

## Acknowledgements

We would like to thank many colleagues, in particular Yuchen Zhang, Jialong Wu, Haoyu Ma, Yuhong Yang, and Jincheng Zhong, for their valuable discussions. This work was supported by the National Key Research and Development Plan (2020AAA0109201), the National Natural Science Foundation of China (62022050 and 62021002), and the Beijing Nova Program (Z201100006820041).

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

# A Algorithm Details

## A.1 ForkMerge

**Proof of Equation** (7).

$$
\begin{aligned}
\lambda^* &= \arg\max_{\lambda} \widehat{\mathcal{P}}(\theta_{t+1}) \\
&= \arg\max_{\lambda} \widehat{\mathcal{P}}(\theta_t - \eta(\mathbf{g}_{\text{tgt}}(\theta_t) + \lambda\mathbf{g}_{\text{aux}}(\theta_t))) \\
&= \arg\max_{\lambda} \widehat{\mathcal{P}}\big((\theta_t - \eta\mathbf{g}_{\text{tgt}}(\theta_t)) + \lambda(-\eta\mathbf{g}_{\text{aux}}(\theta_t))\big) \\
&= \arg\max_{\lambda} \widehat{\mathcal{P}}\big((1-\lambda)(\theta_t - \eta\mathbf{g}_{\text{tgt}}(\theta_t)) + \lambda(\theta_t - \eta(\mathbf{g}_{\text{tgt}}(\theta_t) + \mathbf{g}_{\text{aux}}(\theta_t)))\big) \\
&= \arg\max_{\lambda} \widehat{\mathcal{P}}\big((1-\lambda)\theta_{t+1}(0) + \lambda\theta_{t+1}(1)\big).
\end{aligned}
$$

**Remarks on the search step.**

We provide two search strategies as follows, and we use the first strategy in our experiments.

- *Grid Search*: Exhaustively searching the task-weighting hyper-parameter $\lambda$ through a manually specified subset of the hyper-parameter space, such as $\{0, 0.2, 0.4, 0.6, 0.8, 1.0\}$.
- *Binary Search*: Repeatedly dividing the search interval of $\lambda$ in half and keep the better hyper-parameter.

Random search, bayesian optimization, gradient-based optimization, and other hyper-parameter optimization methods can also be used here, and they are left to be explored in follow-up work.

In practice, the costs of estimating $\widehat{\mathcal{P}}$ in the search step are usually negligible. Yet when the amount of data in the validation set is relatively large, we can sample the validation set to reduce the cost of estimating $\widehat{\mathcal{P}}$.

**Remarks on extension from the one gradient step to $\Delta t$ steps.**

1. It can effectively reduce the average cost of estimating $\widehat{\mathcal{P}}$ at each step and avoid over-fitting the validation set.

2. It allows longer-term rewards from auxiliary tasks and leads to safer task transfer. For instance, when the accumulated gradients of some auxiliary tasks are harmful to the final target performance, the merging step can cancel the effect of these auxiliary tasks by setting their associated weights $\lambda$ to 0, to mitigate strong negative transfer.

3. It increases the risk to produce bad model parameters. However, such risk is still low since deep models usually have smooth loss surfaces after convergence as shown in Section 5.1.

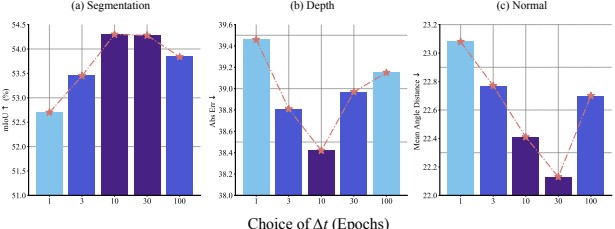

Figure 7: Effect of the merging step $\Delta t$ on NYUv2.

Figure 7 illustrates that an appropriate $\Delta t$ can effectively promote the performance of the ForkMerge algorithm, indicating the necessity of the extension from the one gradient step in previous work to $\Delta t$ steps. When $\Delta t$ is small, the estimation of $\lambda$ is short-insight and might fail to remove the harmful parameter updates when negative transfer occurs, which also indicates the limitations of methods that use single-step gradient descent to estimate $\lambda$ [15, 43]. When $\Delta t$ is large, the risk to get bad model parameters from the linear combination will also increase. Therefore, in our experiment, we use the validation set to pick a proper $\Delta t$ for each dataset and use it for all tasks in this dataset.

## A.2 Use ForkMerge to Select Tasks Simultaneously

**Detailed Algorithm.** Algorithm 2 provides the general optimization process for any task-weighting vector $\{\boldsymbol{\nu}^b\}_{b=1}^B$. For Equation (12), we have $B = K + 1$ and $\boldsymbol{\nu}_i^b = \mathbb{1}[i = b - 1 \text{ or } i = 0]$. For Equation (13), we have no constraints on $B$ or $\boldsymbol{\nu}^b$.

---

**Algorithm 2** ForkMerge Training Pipeline with Multiple Branches

---

**Require:** initial model parameter $\theta_0$, task-weighting vector $\{\boldsymbol{\nu}^b\}_{b=1}^B$, total iterations $T$, interval $\Delta t$
**Ensure:** final model parameter $\theta_T^*$
1: fork model into $B$ copies $\{\theta^b\}_{b=1}^B$
2: **for** $b = 1$ **to** $B$ **do**
3:      $\theta_0^b \leftarrow \theta_0$                                                 ▷ initialization
4: **end for**
5: **while** $t < T$ **do**
6:      **for** $b = 1$ **to** $B$ **do**                                   ▷ independent update
7:          **for** $t' = t$ **to** $t + \Delta t - 1$ **do**
8:              $\theta_{t'+1}^b = \theta_{t'}^b - \eta \sum_k \nu_k^b \mathbf{g}_k(\theta_{t'}^b)$
9:          **end for**
10:      **end for**
11:      $\boldsymbol{\Lambda}^* \leftarrow \arg\max_{\boldsymbol{\Lambda}} \widehat{\mathcal{P}}\big(\sum_b \Lambda_b \theta_{t+\Delta t}^b\big)$
12:                                       ▷ search $\boldsymbol{\Lambda}$ on the validation set
13:      $\theta_{t+\Delta t}^* \leftarrow \sum_b \Lambda_b^* \theta_{t+\Delta t}^b$
14:                                       ▷ merge parameters
15:      **for** $b = 1$ **to** $B$ **do**
16:          $\theta_{t+\Delta t}^b \leftarrow \theta_{t+\Delta t}^*$                            ▷ synchronize parameters
17:      **end for**
18:      $t \leftarrow t + \Delta t$
19: **end while**

---

**Proof of Equation** (12).

The goal of selecting $\boldsymbol{\lambda}^*$ in Equation (10) is to maximize the validation performance of model $\theta_{t+1}$,

$$
\begin{aligned}
\boldsymbol{\lambda}^* &= \arg\max_{\boldsymbol{\lambda}} \widehat{\mathcal{P}}(\theta_{t+1}) \\
&= \arg\max_{\boldsymbol{\lambda}} \widehat{\mathcal{P}}\big(\theta_t - \eta \sum_k \lambda_k \mathbf{g}_k(\theta_t)\big) && //\text{gradient descent} \\
&= \arg\max_{\boldsymbol{\lambda}} \widehat{\mathcal{P}}\big(\theta_t - \eta \lambda_0 \mathbf{g}_0(\theta_t) - \eta \sum_{k \neq 0} \lambda_k \mathbf{g}_k(\theta_t)\big) \\
&= \arg\max_{\boldsymbol{\lambda}} \widehat{\mathcal{P}}\big(\theta_t - \eta \mathbf{g}_0(\theta_t) - \eta \sum_{k \neq 0} \lambda_k \mathbf{g}_k(\theta_t)\big) && //\lambda_0 = 1 \\
&= \arg\max_{\boldsymbol{\lambda}} \widehat{\mathcal{P}}\big((1 - \sum_{k \neq 0} \lambda_k)(\theta_t - \eta \mathbf{g}_0(\theta_t)) + (\sum_{k \neq 0} \lambda_k)(\theta_t - \eta \mathbf{g}_0(\theta_t)) + \sum_{k \neq 0} \lambda_k(-\eta \mathbf{g}_k(\theta_t))\big) && //\sum_{k \neq 0} \lambda_k \leq 1 \\
&= \arg\max_{\boldsymbol{\lambda}} \widehat{\mathcal{P}}\big((1 - \sum_{k \neq 0} \lambda_k)(\theta_t - \eta \mathbf{g}_0(\theta_t)) + \sum_{k \neq 0} \lambda_k(\theta_t - \eta \mathbf{g}_0(\theta_t) - \eta \mathbf{g}_k(\theta_t))\big)
\end{aligned}
$$

By definitions of $\boldsymbol{\Lambda}$ and $\{\boldsymbol{\omega}^k\}_{k=0}^K$

$$
\Lambda_k = \begin{cases} 1 - \sum\limits_{i \neq 0} \lambda_i, & k = 0, \\ \lambda_k, & k \neq 0, \end{cases}
$$

$$
\omega_i^k = \begin{cases} 1, & i = 0 \text{ or } i = k, \\ 0, & \text{otherwise}, \end{cases}
$$

we can prove that optimizing $\boldsymbol{\lambda}$ in Equation (10) is equivalent to optimizing $\boldsymbol{\Lambda}$ as follows:

$$\boldsymbol{\Lambda}^* = \arg\max_{\boldsymbol{\Lambda}} \widehat{\mathcal{P}}\big(\sum_k \Lambda_k \theta_{t+1}(\boldsymbol{\omega}^k)\big).$$

**Remarks on the search step.**

Grid searching all possible values of $\boldsymbol{\Lambda}$ is computationally expensive especially when $\|\boldsymbol{\Lambda}\|$ is large. Thus, here we introduce a greedy search strategy in Algorithm 3, which reduces the computation complexity from exponential complexity to $\mathcal{O}(\|\boldsymbol{\Lambda}\|)$.

---

**Algorithm 3** Greedy Search of $\boldsymbol{\Lambda}^*$

---

**Require:** A list of model parameters $\theta_1, ..., \theta_B$ sorted in decreasing order of $\widehat{\mathcal{P}}(\theta_b)$.
**Ensure:** optimal linear combination coefficient $\boldsymbol{\Lambda}^*$
 1: unnormalized combination coefficient $\widetilde{\boldsymbol{\Lambda}} \leftarrow \mathbf{e}_1$                ▷ initialization
 2: **for** $b = 2$ **to** $B$ **do**
 3:      set upper bound $U \leftarrow \frac{1}{b-1} \sum_{m=1}^{b-1} \widetilde{\Lambda}_m$
 4:      grid search the optimal $\widetilde{\Lambda}_m$ in range $[0, U]$ to maximize $\widehat{\mathcal{P}}(\frac{1}{\|\widetilde{\boldsymbol{\Lambda}}\|} \sum_{m=1}^{b} \widetilde{\Lambda}_m \theta_m)$
 5: **end for**
 6: $\boldsymbol{\Lambda}^* \leftarrow \frac{1}{\|\widetilde{\boldsymbol{\Lambda}}\|} \widetilde{\boldsymbol{\Lambda}}$                  ▷ normalization

---

# B   Analysis Details

In this section, we provide the implementation details of our analysis experiment in Section 3.

We conduct our analysis on the multi-domain image recognition dataset DomainNet [61]. In our analysis, we use task Painting and Quickdraw in DomainNet as examples of weak negative transfer and strong negative transfer, and other tasks (Real, Sketch, Infograph, Clipart) in DomainNet as auxiliary tasks. Details of these tasks are summarized in Table 8. We use ResNet-18 [23] pre-trained on ImageNet [8] for all experiments.

## B.1   Effect of Gradients Conflicts

First, we optimize the model on the target task for $T = 25\text{K}$ iterations to obtain $\theta_T$. We adopt mini-batch SGD with momentum of $0.9$ and batch size of $48$, and the initial learning rate is set as $0.01$ with cosine annealing strategy [45].

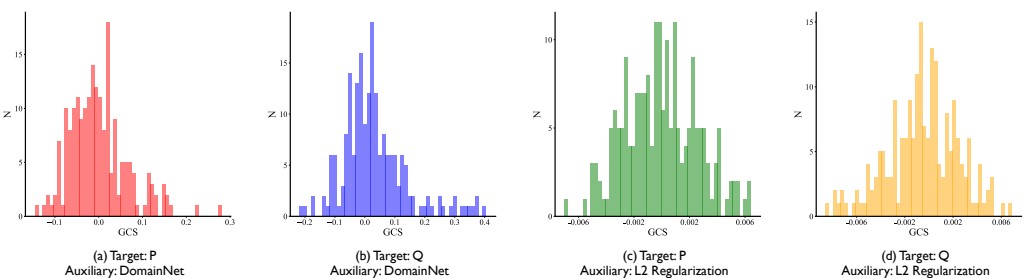

Figure 8: The distribution of Gradient Cosine Similarity (GCS). **P** and **Q** are short for Painting and Quickdraw tasks, respectively.

We repeatedly sample a mini-batch of data and estimate the gradients for the target and auxiliary task $\mathbf{g}_{\text{tgt}}$ and $\mathbf{g}_{\text{aux}}$. Figure 8 plots the distribution of gradient cosine similarity (GCS) between $\mathbf{g}_{\text{tgt}}$ and $\mathbf{g}_{\text{aux}}$. We find that the gradients of different tasks are nearly orthogonal ($\cos \phi_{ij} \approx 0$) in most cases, and highly consistent gradients or severely conflicting gradients are both relatively rare.

Then, we optimize the same $\theta_T$ with one-step multi-task gradient descent estimated from different data to obtain different $\theta_{T+1}$,

$$\theta_{T+1}(\lambda) = \theta_T - \eta(\mathbf{g}_{\text{tgt}}(\theta_T) + \lambda\mathbf{g}_{\text{aux}}(\theta_T)), \tag{14}$$

where $\eta = 0.01$ and $\lambda$ takes values from $\{0, \frac{1}{16}, \frac{1}{8}, \frac{1}{4}, \frac{1}{2}, 1\}$. We evaluate $\theta_{T+1}(\lambda)$ and $\theta_{T+1}(0)$ on the validation set of the target task to calculate the transfer gain (TG) from single-step multi-task gradient descent

$$TG(\lambda) = \widehat{\mathcal{P}}(\theta_{T+1}(\lambda)) - \widehat{\mathcal{P}}(\theta_{T+1}(0)). \tag{15}$$

Note that we omit the notation of algorithm $\mathcal{A}$ in Equation (14) and (15) for simplicity. Then, in Figure 2, we mark the GCS and TG of each data point and fit them with a 3-order polynomial to obtain the corresponding correlation curve.

## B.2 Effect of Distribution Shift

**Qualitative Visualization.** We visualize by t-SNE [74] in Figure 3(a) the representations of the training and test data by the model $\theta_T$ trained in Section B.1. For better visualization, we only keep the top 10 categories with the highest frequency in DomainNet. To visualize the impact of $\lambda$ on the interpolated training distribution, we let the frequency of auxiliary task points be proportional to $\lambda$. In other words, when the weighing hyper-parameter of the auxiliary task increases, the effect of the auxiliary task on the interpolated distribution will also increase.

Figure 3 provides the t-SNE visualization of training and test distributions when $\lambda$ takes values from $\{0, \frac{1}{16}, \frac{1}{4}, 1\}$. We observe that for weak negative transfer tasks, when $\lambda$ initially increases, the area of training distribution can better cover that of the test distribution. But as $\lambda$ continues to increase, the distribution shift between the test set and the training set will gradually increase. For strong negative transfer tasks, however, the shift between the interpolated training distribution and the test distribution monotonically enlarges as $\lambda$ increases.

**Quantitative Measure.** First, we jointly optimize the model on the target task and auxiliary tasks with different weighting hyper-parameter $\lambda$ for $T = 25\text{K}$ iterations to obtain $\theta_T(\lambda)$. We adopt the same hyper-parameters as in Section B.1. Then we evaluate $\theta_T(\lambda)$ on the test set of the target tasks and calculate the average confidence on the test set. We can calculate the confidence score discrepancy (CSD) by Definition 3.5 and the transfer gain (TG) by

$$TG(\lambda) = \widehat{\mathcal{P}}(\theta_T(\lambda)) - \widehat{\mathcal{P}}(\theta_T(0)). \tag{16}$$

Again, we omit the notation of algorithm $\mathcal{A}$ for simplicity. Finally, we plot the curve between CSD and TG under different $\lambda$ in Figure 3(b).

## C Experiment Details

### C.1 Definition of $\Delta_m$

Following [55, 37], we report $\Delta_m$ as the performance measure, which is the average per-task performance improvement of method $m$ relative to the STL baseline $b$. Formally, $\Delta_m = \frac{1}{K}\sum_{k=1}^{K}(-1)^{z_k}(M_{m,k} - M_{b,k})/M_{b,k}$ where $M_{b,k}$ and $M_{m,k}$ is the performance of the $k$-th task obtained by the baseline method $b$ and the compared method $m$. $z_k$ is set to 0 if a higher value indicates better performance for the $k$-th task and otherwise 1.

### C.2 Auxiliary-Task Scene Understanding on NYU

**Experiment Details.** We use DeepLabV3+ architecture [1], where a ResNet-50 network [23] pretrained on the ImageNet dataset [8] with dilated convolutions is used as a shared encoder among tasks and the Atrous Spatial Pyramid Pooling module is used as task-specific head for each task. Following [44, 79], each method is trained for 200 epochs with the Adam optimizer [29] and batch size of 8. The initial learning rate is $10^{-4}$ and halved to $5 \times 10^{-5}$ after 100 epochs. In ForkMerge, the parameters are merged every 10 epochs. Table 7 presents the full evaluation results of Table 1.

Table 7: Performance on NYUv2 dataset.

| Methods | Segmentation | | Depth | | Normal | | | | | | $\mathbf{\Delta}_m\uparrow$ |
|---|---|---|---|---|---|---|---|---|---|---|---|
| | | | | | Angle Distance | | Within $t°$ | | | | |
| | mIoU↑ | Pix Acc↑ | Abs Err↓ | Rel Err↓ | Mean↓ | Median↓ | 11.25↑ | 22.5↑ | 30↑ | |
| STL | 51.42 | 74.14 | 41.74 | 17.37 | 22.82 | 16.23 | 36.58 | 62.75 | 73.52 | - |
| EW | 52.13 | 74.51 | 39.03 | 16.43 | 24.14 | 17.62 | 33.98 | 59.63 | 70.93 | 0.30% |
| UW | 52.51 | 74.72 | 39.15 | 16.56 | 23.99 | 17.36 | 34.46 | 60.13 | 71.32 | 0.63% |
| DWA | 52.10 | 74.45 | 39.26 | 16.57 | 24.12 | 17.62 | 33.88 | 59.72 | 71.08 | 0.07% |
| RLW | 52.88 | 74.99 | 39.75 | 16.67 | 23.83 | 17.23 | 34.76 | 60.42 | 71.50 | 0.66% |
| MGDA | 50.79 | 73.81 | 39.19 | 16.25 | 23.14 | 16.46 | 36.15 | 62.17 | 72.97 | 1.44% |
| GradNorm | 52.25 | 74.54 | 39.31 | 16.37 | 23.86 | 17.46 | 34.13 | 60.09 | 71.45 | 0.56% |
| PCGrad | 51.77 | 74.72 | 38.91 | 16.36 | 24.31 | 17.66 | 33.93 | 59.43 | 70.62 | 0.22% |
| IMTL | 52.24 | 74.73 | 39.46 | **15.92** | 23.25 | 16.64 | 35.86 | 61.81 | 72.73 | 2.10% |
| GradVac | 52.84 | 74.77 | 39.48 | 16.28 | 24.00 | 17.49 | 34.21 | 59.94 | 71.26 | 0.75% |
| CAGrad | 52.04 | 74.25 | 39.06 | 16.30 | 23.39 | 16.89 | 35.35 | 61.28 | 72.42 | 1.41% |
| NashMTL | 51.73 | 74.10 | 39.55 | 16.50 | 23.21 | 16.74 | 35.39 | 61.80 | 72.92 | 1.11% |
| GCS | 52.67 | 74.59 | 39.72 | 16.64 | 24.10 | 17.56 | 34.04 | 59.80 | 71.04 | 0.09% |
| OL_AUX | 52.07 | 74.28 | 39.32 | 16.30 | 23.98 | 17.87 | 33.89 | 59.53 | 71.08 | 0.17% |
| ARML | 52.73 | 74.85 | 39.61 | 16.65 | 23.89 | 17.50 | 34.24 | 59.87 | 71.39 | 0.37% |
| Auto-$\lambda$ | 52.40 | 74.62 | 39.25 | 16.25 | 23.38 | 17.20 | 34.05 | 61.18 | 72.05 | 1.17% |
| Post-train | 52.08 | 74.86 | 39.58 | 16.77 | 22.98 | 16.48 | 36.04 | 62.27 | 73.20 | 1.49% |
| ForkMerge | **53.67** | **75.64** | **38.91** | 16.47 | **22.18** | **15.60** | **37.93** | **64.29** | **74.81** | **4.03%** |

## C.3 Auxiliary-Domain Image Recognition on DomainNet

**Dataset Details.** As the original DomainNet [61] does not provide a separate validation set, we randomly split $50\%$ data from the test set as the validation set, and use the rest $50\%$ data as the test set. For each task, the proportions of training set, validation set, and test set are approximately $70\%/15\%/15\%$. Table 8 summarizes the statistics of this dataset. DomainNet is under Custom (research-only, non-commercial) license.

Table 8: Overview of DomainNet dataset.

| Tasks | #Train | #Val | #Test | Description |
|---|---|---|---|---|
| Clipart | 33.5K | 7.3K | 7.3K | collection of clipart images |
| Real | 120.9K | 26.0K | 26.0K | photos and real world images |
| Sketch | 48.2K | 10.5K | 10.5K | sketches of specific objects |
| Infograph | 36.0K | 7.8K | 7.8K | infographic images |
| Painting | 50.4K | 10.9K | 10.9K | painting depictions of objects |
| Quickdraw | 120.7K | 25.9K | 25.9K | drawings of game "Quick Draw" |

**Experiment Details.** We adopt mini-batch SGD with momentum of 0.9 and batch size of 48. We search the initial learning rate in $\{0.003, 0.01, 0.03\}$ and adopt cosine annealing strategy [45] to adjust learning rate during training. We adopt ResNet-101 pretrained on ImageNet as the backbone. Each method is trained for 50K iterations. In ForkMerge, the parameters are merged every 12.5K iterations.

## C.4 CTR and CTCVR Prediction on AliExpress

**Dataset Details.** AliExpress [36] is gathered from the real-world traffic logs of AliExpress search system in Taobao and contains more than 100M records in total. We split the first $90\%$ data in the time sequence to be training set and the rest $5\%$ and $5\%$ to be validation set and test set. AliExpress consists of 2 tasks: click-through rate (CTR) and click-through conversion rate (CTCVR), and 4 scenarios: Spain (ES), French (FR), Netherlands (NL), and America (US). Table 9 summarizes the statistics of this dataset. AliExpress is under Attribution-NonCommercial-ShareAlike 4.0 International (CC BY-NC-SA 4.0) license.

**Experiment Details.** The architecture of most methods is based on ESMM [47], which consists of a single embedding layer shared by all tasks and multiple independent DNN towers for each task. The embedding dimension for each feature field is 128. Each method is trained for 50 epochs using the Adam optimizer, with the batch size of 2048, learning rate of $10^{-3}$ and weight decay of $10^{-6}$.

Table 9: Overview of AliExpress dataset, where CTR = #Click / #Impression and CTCVR = #Purchase / #Impression.

| Statistics | ES | FR | NL | US |
|---|---|---|---|---|
| #Product | 8.7M | 7.4M | 6M | 8M |
| #Pv | 2M | 1.7M | 1.2M | 1.8M |
| #Impression | 31.6M | 27.4M | 17.7M | 27.4M |
| #Click | 841K | 535K | 382K | 450K |
| #Purchase | 19.1K | 14.4K | 13.8K | 10.9K |
| CTR | 2.66% | 2.01% | 2.16% | 1.64% |
| CTCVR | 0.60‰ | 0.54‰ | 0.78‰ | 0.40‰ |

### C.5 Semi-supervised Learning on CIFAR10 and SVHN

**Dataset Details.** Following [67], we first split the original training set of CIFAR10 [31] and SVHN [56] into training set and validation set. Then, we randomly sample labeled images from the training set. Table 10 summarizes the statistics of CIFAR-10 and SVHN.

Table 10: Overview of CIFAR-10 and SVHN datasets.

| Datasets | #Labeled | #Unlabeled | #Val | #Test |
|---|---|---|---|---|
| CIFAR-10 | 4000 | 41000 | 5000 | 10000 |
| SVHN | 1000 | 64931 | 7326 | 26032 |

**Experiment Details.** (1) **Auxiliary Tasks.** Following [82, 67], we consider two self-supervised auxiliary tasks Rotation [19] and Exempler-MT [14]. In Rotation, we rotate each image by $[0°, 90°, 180°, 270°]$ and ask the network to predict the angle. In Exemplar-MT, the model is trained to extract feature invariant to a wide range of image transformations. (2) **Hyper-parameters.** We adopt Adam [29] optimizer with an initial learning rate of $0.005$. We train each method for $200K$ iterations and decay the learning rate by a factor of $0.2$ at $160K$ iterations. We use Wide ResNet-28-2 [80] as the backbone. In ForkMerge, the parameters are merged every $10K$ iterations.

### C.6 Data Division Strategy for ForkMerge

As discussed in Section 4.2, in ForkMerge, we can construct branches with different sets of auxiliary tasks. Below we outline the specific data division strategy used in our experiments, which is consistent with previous ATL literature:

- For the NYUv2 dataset, multiple tasks share the same input, but their outputs are different. In this setup, each branch has the same input data, which includes the entire dataset. The distinction between different branches solely lies in the task weighting vector $\{\boldsymbol{\nu}^b\}_{b=1}^B$.

- For DomainNet, AliExpress, CIFAR-10, and SVHN datasets, different tasks have both different inputs and outputs. In these cases, for each branch, if the task weighting of a specific task is set to 0, the data from that particular task will not be used for training the corresponding branch.

## D   Additional Experiments

### D.1   Analysis on the importance of different forking branches

**The importance of different forking branches is dynamic.** As shown in Figure 9, the relative ratio of each forking branch is dynamic and varies from task to task, which indicates the importance of the dynamic merge mechanism.

### D.2   Analysis on the computation cost

The computation cost of Algorithm 2 is $\mathcal{O}(K)$ and the computation cost of the pruned version is $\mathcal{O}(B)$. Usually, only one model is optimized in most previous multi-task learning methods, yet their

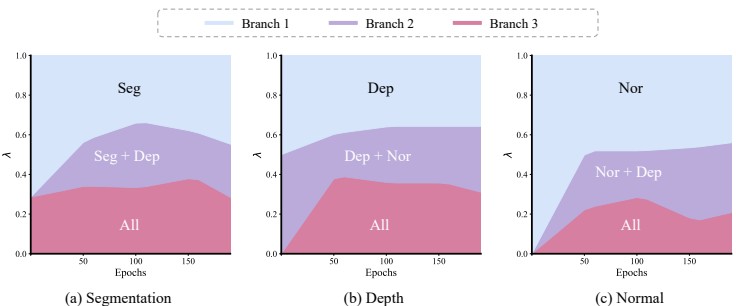

Figure 9: Importance of different forking branches during training on NYUv2.

computational costs are not necessarily $\mathcal{O}(1)$. Gradient balancing methods, including MGDA [65], GradNorm [5], PCGrad [79], IMTL [41], GradVac [76], CAGrad [40], NashMTL [55], GCS [15], OL_AUX [39], and ARML [67], require computing gradients of each task, thus leading to $\mathcal{O}(K)$ complexity. In addition, calculating the inner product or norm of the gradients will bring a calculation cost proportional to the number of network parameters. A common practical improvement is to compute gradients of the shared representation [65]. Yet the speedup is architecture-dependent, and this technique may degrade performance [55].

In Figure 10, we also compare the actual training time across these methods on NYUv2. We can observe that ForkMerge does not require more time than most other methods. And considering the significant performance gains it brings, these additional computational costs are also worth it. Furthermore, our fork and merge mechanism enables extremely easy asynchronous optimization which is not straightforward in previous methods, thus the training time of our method can be reduced to $\mathcal{O}(1)$ when there are multiple GPUs available.

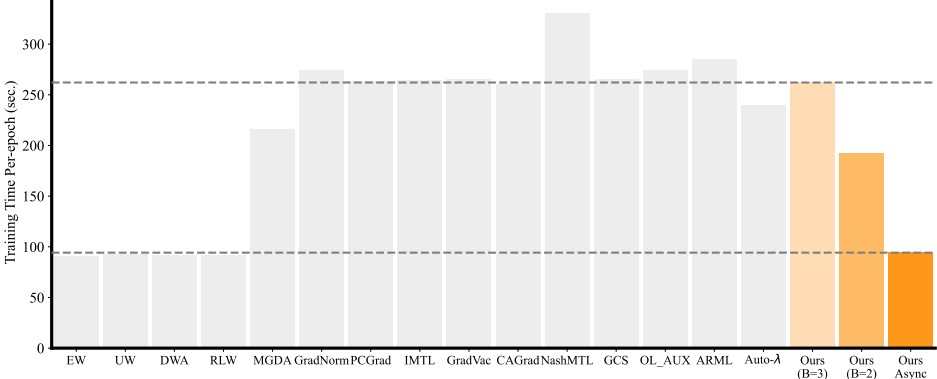

Figure 10: Training speed of different MTL methods on NYUv2 (10 repetitions).

### D.3 Analysis on the convergence and variance

Figure 11 plots the validation performance of STL, EW, and ForkMerge throughout the training process on NYUv2. Each curve is obtained by optimizing the same method with 5 different seeds. Compared with single-task learning or minimizing the average loss across all tasks, ForkMerge not only improves the final generalization but also speeds up the convergence and reduces the fluctuations during training.

### D.4 Comparison with grid searching $\lambda$

In Section 3, we observe that adjusting the task-weighting hyper-parameter $\lambda$ can effectively reduce the negative transfer and promote the positive transfer. [77] also suggests that sweeping the task weights should be sufficient for full exploration of the Pareto frontier at least for convex setups and observe no improvements in terms of final performance from previous MTL algorithms compared with grid search.

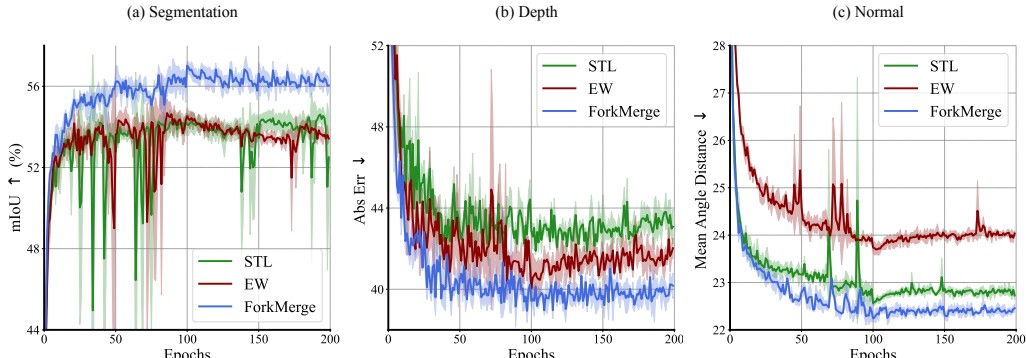

Figure 11: Learning curves comparing different methods on NYUv2. Each curve plots the mean and standard deviation of the validation performance of a method with 5 different random seeds.

In Figure 12, we compare the performance of all methods with grid search on NYUv2. In grid search, the weighting hyper-parameter for each task takes values from $\{0.3, 1.0, 3.0\}$, and there are 3 tasks in NYUv2, thus there are 27 combinations in total. We find that previous methods simply yield performance trade-off points on the scalarization Pareto front, which has also been observed in previous work [77]. In contrast, our proposed ForkMerge yields point far away from the Pareto front and achieves significant improvement over simply optimizing a weighted average of the losses. One possible reason for the gain is that the task weighting in grid search is fixed during training and takes finite values due to the limitation of computing resources while the task weighting in ForkMerge is *dynamic* in time and nearly *continuous* in values, thus can better avoid negative transfer and promote positive transfer.

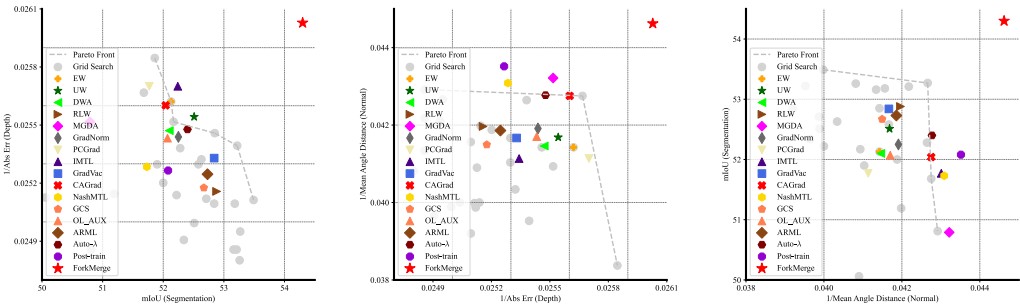

Figure 12: Comparison with grid search on NYUv2. We use mIoU for semantic segmentation, $1/$ absolute error for depth estimation, and $1/$ mean angle distance for surface normal prediction. We plot 2D projections of the performance profile for each pair of tasks. Top-right is better.

### D.5 Comparison with larger batch size training

In some sense, the multiple branches in ForkMerge increase the equivalent batch size. And it has been revealed that batch size may have a great effect on the performance of deep models [21, 78]. To ablate the influence of batch size, we increase the batch size of the Equal Weighting method. As shown in Table 11, the improvement brought by ForkMerge itself is significantly larger than simply increasing the batch size.

### D.6 ForkMerge with more network architectures

**ForkMerge with Vision Transformers.** We replace the backbone network ResNet-101 with advanced ViT-Base [13] pretrained on ImageNet-21K and repeat the experiments on the DomainNet dataset (Section 5.2). As demonstrated in Table 12, when employing the Vision Transformer model, which boasts increased capacity, the risk of overfitting with limited data becomes more pronounced.

Table 11: Comparison of different methods with larger batch size training.

| Methods | Batch Size | Segmentation | | Depth | | Normal | | | | | | $\Delta_m\uparrow$ |
|---------|-----------|------|------|------|------|------|------|------|------|------|------|------|
| | | | | | | Angle Distance | | Within $t^\circ$ | | | | |
| | | mIoU↑ | Pix Acc↑ | Abs Err↓ | Rel Err↓ | Mean↓ | Median↓ | 11.25↑ | 22.5↑ | 30↑ | |
| EW | 8 | 52.13 | 74.51 | 39.03 | 16.43 | 24.14 | 17.62 | 33.98 | 59.63 | 70.93 | 0.30% |
| EW | 32 | 51.40 | 73.99 | 38.86 | 16.20 | 23.99 | 17.34 | 34.58 | 60.08 | 70.85 | 0.55% |
| ForkMerge‡ | 8 | **54.30** | **75.78** | **38.42** | **16.11** | **22.41** | **15.72** | **37.81** | **63.89** | **74.35** | **4.59%** |

This makes Single Task Learning (STL) less effective and consequently leads to the Equal Weighting (EW) method outperforming STL, causing the Post-train method to fall short of EW and Auto-$\lambda$. In this case, ForkMerge still exhibited superior performance, validating its efficacy across different network architectures.

Table 12: Performance on DomainNet by replacing the ResNet-101 architecture with the ViT-Base architecture.

| Methods | C | I | P | Q | R | S | Avg | $\Delta_m\uparrow$ |
|---------|------|------|------|------|------|------|------|------|
| STL | 75.7 | 37.8 | 69.0 | 72.1 | 84.4 | 69.0 | 68.0 | - |
| EW | 81.9 | 43.7 | 74.0 | 71.3 | 84.1 | 73.0 | 71.3 | +5.90% |
| Auto-$\lambda$ | 81.3 | 44.1 | 73.8 | 72.1 | 84.4 | 73.5 | 71.5 | +6.62% |
| Post-train | 76.2 | 38.8 | 69.5 | 71.7 | 83.2 | 69.7 | 68.2 | +0.51% |
| ForkMerge | **83.0** | **45.6** | **76.3** | **73.2** | **87.1** | **74.7** | **73.3** | **+8.97%** |

**ForkMerge with Multi-task Architectures.** ForkMerge is complementary to different multi-task architectures. In Tables 13 and 14, we provide a comparison of different optimization strategies with MTAN [44] and MMoE [46] as architectures, which are widely used in multi-task computer vision tasks and multi-task recommendation tasks respectively. On these specifically designed multi-task architectures, ForkMerge is still significantly better than other methods.

Table 13: Performance on NYUv2 dataset by replacing the DeepLabV3+ architecture with the MTAN architecture.

| Methods | Segmentation | | Depth | | Normal | | | | | | $\Delta_m\uparrow$ |
|---------|------|------|------|------|------|------|------|------|------|------|------|
| | | | | | Angle Distance | | Within $t^\circ$ | | | | |
| | mIoU↑ | Pix Acc↑ | Abs Err↓ | Rel Err↓ | Mean↓ | Median↓ | 11.25↑ | 22.5↑ | 30↑ | |
| STL | 52.10 | 74.42 | 40.45 | 16.34 | 22.35 | 15.23 | 38.96 | 64.56 | 74.51 | 3.05% |
| EW | 53.27 | 75.36 | 39.37 | 16.38 | 23.61 | 17.00 | 35.00 | 61.01 | 72.07 | 1.62% |
| GCS | 53.05 | 74.79 | 39.50 | 16.49 | 24.05 | 17.49 | 34.14 | 59.88 | 71.13 | 0.57% |
| OL_AUX | 52.47 | 74.70 | 39.27 | 16.39 | 23.66 | 17.43 | 34.49 | 59.96 | 71.76 | 0.82% |
| ARML | 52.33 | 74.59 | 39.46 | 16.61 | 23.57 | 17.41 | 34.56 | 60.12 | 72.04 | 0.55% |
| Auto-$\lambda$ | 52.90 | 75.03 | 39.67 | 16.45 | 22.71 | 15.60 | 38.35 | 64.09 | 73.92 | 3.18% |
| ForkMerge‡ | **55.25** | **76.16** | **38.45** | **16.08** | **21.94** | **15.22** | **38.96** | **65.04** | **75.33** | **5.76%** |

Table 14: Performance on AliExpress dataset by replacing the ESMM architecture with the MMoE architecture.

| Methods | CTR | | | | CVCTR | | | | Avg | $\Delta_m\uparrow$ |
|---------|------|------|------|------|------|------|------|------|------|------|
| | ES | FR | NL | US | ES | FR | NL | US | | |
| EW | 0.7287 | 0.7244 | 0.7225 | 0.7068 | 0.8874 | 0.8669 | 0.8688 | 0.8742 | 0.7974 | 0.11% |
| GCS | 0.7300 | 0.7190 | 0.7270 | 0.7102 | 0.8857 | 0.8773 | 0.8680 | 0.8740 | 0.7989 | 0.29% |
| OL_AUX | 0.7265 | 0.7283 | 0.7264 | **0.7146** | 0.8849 | 0.8750 | 0.8710 | 0.8770 | 0.8005 | 0.50% |
| ARML | 0.7289 | 0.7278 | 0.7248 | 0.7081 | 0.8869 | 0.8801 | 0.8714 | 0.8610 | 0.7986 | 0.26% |
| Auto-$\lambda$ | 0.7269 | 0.7273 | 0.7256 | 0.7111 | 0.8827 | **0.8811** | **0.8721** | 0.8726 | 0.7999 | 0.42% |
| ForkMerge‡ | **0.7368** | **0.7349** | **0.7359** | 0.7116 | **0.8942** | 0.8791 | 0.8717 | **0.8840** | **0.8060** | **1.20%** |

