# OpenReview forum: "ForkMerge: Mitigating Negative Transfer in Auxiliary-Task Learning"
_NeurIPS.cc/2023/Conference — NeurIPS 2023 poster_

### Official Review · Reviewer_ECtn · 2023-07-07

**Soundness:** 3 good
**Presentation:** 3 good
**Contribution:** 3 good
**Rating:** 7
**Confidence:** 5

**Summary:**

This paper considers how to best use auxiliary tasks to improve performance on target tasks. Specifically, a "ForkMerge" procedure is proposed which consists of two parallel optimization procedures, one on the target task, and one which includes auxiliary data, and the resulting weights are synchronized at regular intervals. Compared to grid searching for an appropriate interpolation factor, the proposed approach can dynamically alter the interpolation between the two sets of model parameters. Experiments on diverse set of benchmarks and a wide array of baselines show that the proposed approach is effective at improving performance using auxiliary tasks, relative to the state of the art.

**Strengths:**

* This paper tackles a difficult problem of improving target task performance using auxiliary data.

* The approach is well-motivated and the experiments are quite convincing.

* The justification in terms of reducing distribution shift relative to test data is quite interesting.

**Weaknesses:**

* The task selection with multiple auxiliary tasks is computationally expensive, and the impact of the pruning procedure is not completely clear.

* There is no adequate discussion of limitations.

**Questions:**

None.

**Limitations:**

See "Weaknesses"

---

> ### Author Rebuttal · Authors · 2023-08-09
>
> We would like to sincerely thank Reviewer ECtn for providing insightful reviews and valuable comments. We have clarified the questions in the following response.
>
> **Q1:** Impact of pruning strategy.
>
> $\text{Table 4}$ illustrates the impact of the pruning strategy of ForkMerge. As the number of branches increases, the cumulative benefits derived from auxiliary tasks become more pronounced. However, while the overall gain increases with the number of branches, the individual gain per branch tends to diminish. This phenomenon reveals the trade-off between computational efficiency and task performance. And our pruning strategy is an approach that allows users to customize the number of branches based on their specific needs and available computational resources.
>
> **Q2:** Computational efficiency and limitation.
>
> We acknowledge the computational demands of task selection involving multiple auxiliary tasks. Indeed, handling such complexity is a challenge that warrants attention. We view the integration of previous task grouping methods [1-3] with our auxiliary task learning approach as a promising avenue for further research. By combining task selection techniques with our method, we envision the potential to alleviate the computational burden while still harnessing the power of auxiliary tasks for improved auxiliary-task learning performance.
>
> [1] *Christopher Fifty, Ehsan Amid, Zhe Zhao, Tianhe Yu, Rohan Anil, and Chelsea Finn. Efficiently identifying task groupings for multi-task learning. In NeurIPS, 2021.*
>
> [2] *Trevor Standley, Amir Zamir, Dawn Chen, Leonidas J. Guibas, Jitendra Malik, and Silvio Savarese. Which tasks should be learned together in multi-task learning? In ICML, 2020.*
>
> [3] *Amir Roshan Zamir, Alexander Sax, William B. Shen, Leonidas J. Guibas, Jitendra Malik, and Silvio Savarese. Taskonomy: Disentangling task transfer learning. In CVPR, 2018.*

---

### Official Review · Reviewer_pM6c · 2023-07-07

**Soundness:** 3 good
**Presentation:** 3 good
**Contribution:** 3 good
**Rating:** 6
**Confidence:** 4

**Summary:**

The paper tackles the problem of learning multiple tasks together which is known to lead to "task inteference" or "negative transfer" issues. This is usually tackled by automatically scaling the task weights or gradients based on training statistics (e.g. GradNorm or uncertainty weighing of losses). In particular, the paper studies an asymetric version of the problem, Auxiliary Task Learning (**ATL**) where one task may not be important at inference but can greatly improve performance when jointly trained with the main target task.

The paper first explore potential causes of task interference under the lens of train/target distribution shifts. Then, the paper proposes **ForkMerge**, a novel optimization method to avoid task interference: For each update of ForkMerge, first the current parameters are duplicated. Then, the weights  are updated separately; one on the target task, while the other is jointly trained for the task and auxiliary tasks. Then, the optimal task weights are found by finding the linear combination of these two sets of weights that maximize validation accuracy (**Equation 12**). The algorithm can be extended to multiple auxiliary tasks which requires an additional separate optimization branch for each new auxiliary task.

The method is then evaluated on the NYU dataset (3 tasks) and DomainNet (6 domains) and compared to previous MTL and ATL approaches

**Strengths:**

- I liked the analysis section of the paper and it contains interesting insights on the problem of task interference: For instance, gradient conflicts is often named as a cause of negative transfer but its effect/strength is rarely actually measured in practice. Although it would have been interesting to extend the analysis to more diverse scenarios.

- I also like the insights that task weighing methods should take generalization into account, which is captured in the proposed algorithm by finding the optimal task weights via evaluation on the validation set

- The paper presents extensive experiments on two classical multitask/multidomain benchmarks, which are further completed in the supplementary material (e.g. additional backbone)

**Weaknesses:**

- **Training cost**: the paper should discuss the training cost in terms of memory efficiency more concretely/quantitatively: ForkMerge requires a separate set of parameters for each task. In addition these parameters are also updated independently hence the number of forward/backward passes will also increases with t he number of tasks. Finally, the cost of search for optimal $\Lambda$ in Equation 13 will also increase with the number of tasks. These costs may remain reasonable in the application scenario with only one auxiliary task, but it's not clear how practical ForkMerge becomes when dealing with more than two tasks like in Section 5.2

- **Finding 2 does not seem very significant**. Finding 2 states that negative transfer is likely to occur when the added auxiliary task increases the train/test distribution shift for the target task: This seems like a pretty classic statement from statistical machine learning theory: For instance generalization bounds for domain adaptation typically include a term measuring the discrepancy between the source / target domains.

- **Chosen baselines might not all be a fair choice**:  the multi-task optimization baselines (MGDA, GradNorm, PCGrad... etc) typically aim to optimize performance on **all tasks**, as opposed to auxiliary task learning where there is a clear bias towards the target tasks: For instance in ForkMerge, every branch includes the target task objective. For that reason, I do not think applying MTL methods "as-is" to the problem of ATL is the best baseline. A stronger baseline could be to finetune the task weights in the "Equal Weights" (**EW**) baseline for performance on the target task; this might be to costly for DomainNet but would be reasonable for NYU as there are only two auxilair ytasks.

**Questions:**

- **Suggestion of writing**: I feel like the paper sometimes overuse abbreviations. For instance only mentioning the tasks **P** and **Q** for DomainNet (Painting and Quickdraw domains ?) assumes the reader is very familiar with the dataset. In additional, there are many abbreviations introduced in the paper which often make the text hard to read (for instance lines 154-159)

- **Distribution shift and hypothesis shift**: I did not understand the point that *lines 213-221* try to convey and the difference between Equations 12 and 13 (outside of renaming $K$ to $B$ and allowing $\Delta_t$ time steps instead of only 1). Similarly, I'm not sure what insight  **Figure 6**  tries to convey. More specifically: the task weighting $\lambda$ is initially defined at the loss level (Equation 1); However, it seems that in later sections $\lambda$ is rephrased as a parameter mixing the distribution of the target and auxiliary tasks; while there are links between these two views, I found that switching between the two interpretations in the text a bit confusing. It is also not clear what it means for multi-task applications: There is only one input distribution (the input images are the same for all tasks) so which distribution shift are we referring to ?

- From the appendix **C.3**, it seems that the test set of DomainNet was split into two parts: one for validation, and the other one for test performance. I think this should be clearly stated in the main paper directly as it means the reported results are not comparable to other DomainNet works

**Limitations:**

There is no dedicated limitations section but the paper discusses the limitation of dedicating independent optimization branches for every auxiliary task (e.g. branch pruning to avoid handling too many branches in the DomainNet)

---

> ### Author Rebuttal · Authors · 2023-08-09
>
> We would like to sincerely thank Reviewer pM6c for providing insightful reviews and valuable comments. We have clarified the questions in the following response.
>
> **Q1:** Concern on the training cost.
>
> Please refer to $\text{question 2 (Q2)}$ of our global rebuttal.
>
> **Q2:** Contribution of Finding 2.
>
> It is worth emphasizing that our contribution lies in providing a new perspective, that is, to consider the problem of auxiliary task learning from the standpoint of model generalization, and to offer a quantifiable metric in this context.
>
> Indeed, traditional measures of distribution distance, such as the Maximum Mean Discrepancy [1] and  $\mathcal{H}\Delta\mathcal{H}$-Divergence [2], have been extensively employed in well-established problems like domain adaptation. However, their direct application to the realm of auxiliary task learning encounters specific challenges.
>
> The primary challenge stems from the original definition of Maximum Mean Discrepancy and $\mathcal{H}\Delta\mathcal{H}$-Divergence, which are based on marginal distributions. To overcome this limitation, we introduce an innovative approach by establishing distribution shift considerations in the output space, leading to $\text{Definition 3.5}$.
>
> [1] *Arthur Gretton, Karsten M. Borgwardt, Malte J. Rasch, Bernhard Sch olkopf, and Alexander Smola. A kernel two-sample test. In JMLR, 2012.*
>
> [2] *Ben-David, S., Blitzer, J., Crammer, K., and Pereira, F. Analysis of representations for domain adaptation. In NeurIPS, 2007.*
>
> **Q3:** Comparison with stronger baseline.
>
> We follow the reviewer's suggestion and enhance the performance of baselines MGDA, GradNorm, and PCGrad by fine-tuning the model obtained from multitask learning to each task. $\text{Table 4 of our global rebuttal pdf}$ presents the results on the NYUv2 dataset, where ForkMerge consistently outperforms improved baselines.
>
> **Q4:** Suggestion of writing.
>
> Many thanks to the reviewer's feedback. As revision is not supported in the rebuttal stage, we will update our camera-ready version as follows:
>
> - We will reduce the use of abbreviations. For instance, in the analysis part ($\text{Section 3}$), we will directly use the full concepts such as "Transfer Gain," "Weak Negative Transfer," "Strong Negative Transfer," and "Confidence Score Discrepancy" instead of their abbreviations.
>  - Besides, we will include a table that explains the meaning of all abbreviations used in the paper.
>
> **Q5:**  Confusion on the distribution shift and hypothesis shift.
>
> **Explanation about $\lambda$.**
> In our paper, $\lambda$ serves a dual role—it acts as both the task weighting parameter at the loss level and as a coefficient governing the mixture of distributions associated with different tasks.
>
> - As discussed in $\text{Section 3.2}$, adjusting the task weighting parameter λ leads to a change in the data distribution that the model is adapting to. This occurs due to the fact that varying λ influences the emphasis placed on different tasks during the learning process. We formally define the interpolated distribution in $\text{Equation 3}$.
> - The equivalence arises from the fact that the model trained on the interpolated distribution $\mathcal{T}\_\text{inter}$ is essentially the same as the model obtained through auxiliary-task learning across the distributions $\mathcal{T}\_\text{tgt}$ and $\mathcal{T}\_\text{aux}$.
>
> **Explanation about distribution.**
> We emphasize that when referring to distribution, we are specifically addressing joint distribution. This is crucial, as while auxiliary tasks might share the same input space, their output spaces invariably differ due to the inherent nature of diverse tasks.
>
> **Explanation about Figure 6.**
>
> - The mixture of hypotheses is different from the mixture of distributions.  To maintain this distinction, we consistently employ the symbol $\lambda$ to represent the mixture weights of distributions, while employing $\Lambda$ to denote the mixture weights of model hypotheses.
> - Regarding $\text{Figure 6}$, it intends to convey the similarity in outcomes between the mixture of model hypotheses and the mixture of distributions. This alignment in behavior enables us to assert that the mixture of hypotheses is an approximation of the mixture of distributions.
>
> **The difference between Equations 12 and 13.**
>
> $\text{Equation 13}$ represents a generalized form of $\text{Equation 12}$, and its generality can be attributed to three key aspects:
>
> - **Customizable Branching**: Unlike $\text{Equation 12}$ which involves a fixed number of tasks, $\text{Equation 13}$ allows for a more flexible setting with a user-defined branching number $B$, which allows for scenarios where pruning or modifying the original candidate branches is desired.
>
> - **Enhanced Flexibility in Candidate Branches**: Additionally, $\text{Equation 13}$ offers a broader perspective on candidate branches. It's not strictly constrained to target-auxiliary task pairs for optimization. Instead, the formulation is open to incorporating any desired custom design of candidate branches, accommodating the integration of prior knowledge or domain-specific insights.
>
> - **Extended Time for Lambda Estimation**: Importantly, $\text{Equation 13}$ considers the changes in estimating the optimal lambda ($\lambda$) by extending the time for estimation. We have noticed that methods like Auto-$\lambda$, which try to estimate $\lambda$ within a single gradient update, often face challenges due to estimation fluctuations. ForkMerge addresses this by using a longer time for $\lambda$ estimation, which reduces the negative impact of estimation errors on model parameters and makes the algorithm more robust.
>
> **Q5**: Split of DomainNet.
>
> Thanks for the feedback. In our camera-ready version, we will explicitly highlight this split difference in the experimental section.

---

> > ### Comment · Reviewer_pM6c · 2023-08-20
> > **thanks for your reply**
> >
> > Hello authors, thanks for your reply and clarifications! Regarding the difference between distribution and hypothesis shift, I still think the way it is described in the paper introduces more confusion than necessary (which ties in with the weakness I listed about finding 2) although it is more clear now thanks to your reply. I would like to keep my original rating of (6).
> >
> > **Minor comments, mainly about writing clarity**
> >
> > **a. About the difference between $\lambda$ and $\Lambda$**
> > In the response, you explain that  *"we consistently employ the symbol to represent the mixture weights of distributions, while employing to denote the mixture weights of model hypotheses"*; however $\lambda$ is initially defined as the weights in the task losses: Building on this, Equation 11 only defines the vector $\Lambda$ to be  essentially equal to $\lambda$; similarly Equation (7) and (8) both define mixture of weights using $\lambda$ and not $\Lambda$.
> >
> > Going further, it seems that instead, the intuition of $\lambda$ as distribution mixture coefficient is the one that should be redefined ? For instance, in lines 216, you write *Thus, we transform the problem of mixture distribution into that of mixture hypothesis*. However the previous paragraph(s) only show how yo transform the problem of mixture of losses to that of mixture hypothesis (equations 9 to 12): As a reader, I found that the link between "mixture of losses" and "mixture of distribution" should be better formalized/motivated; it's briefly mentioned in Section 3.2 (equation 3) but does not really tie in with the rest of the paper and the method of Section 4
> >
> > **b. regarding Equation 13**,
> > I understand the points you mentioned, but I think the writing should be rephrased to motivate Equation (13) better: For instance in the paper, $B$ is only introduced as "number of candidate branches" while in your response, you clearly mention its use for pruning/efficiency. The **extended time for lambda estimation** is not mentioned at all when introducing (13) (but it makes sense from a practical perspective). And similarly, the **enhanced flexibility** wrt to Equation 12 is also not mentioned, and is not apparent from the equation itself.

---

> > > ### Author Response · Authors · 2023-08-21
> > > **Thanks for the Reviewer's Reply**
> > >
> > > We'd like to thank Reviewer pM6c again for providing an impressively insightful pre-rebuttal review, which has enabled us to make an effective response. We'd also thank you for carefully judging our feedback and acknowledging our work in the final review.
> > >
> > > Following your suggestion on writing clarity, in the next version, we will delve deeper into the relationship between the concepts of "mixture of losses" and "mixture of distributions." We will also provide more explanation on $\text{Equation 13}$ to highlight its distinctions and advantages in comparison to $\text{Equation 12}$.

---

### Official Review · Reviewer_uS5H · 2023-07-23

**Soundness:** 3 good
**Presentation:** 2 fair
**Contribution:** 3 good
**Rating:** 7
**Confidence:** 3

**Summary:**

The authors conduct an analysis of negative transfer in auxiliary task learning, finding that gradient conflicts are not necessarily tied to negative transfer, but that auxiliary tasks that induce large distribution shifts from the new training distribution to the test distribution tend to cause negative transfer. The authors then propose ForkMerge, which repeatedly forks a model into branches trained on just the target task and trained on both the target task and auxiliary task(s), and then uses target task validation set performance to determine how to merge the forked models.

**Strengths:**

1. The authors share an interesting and thoughtful analysis, investigating both gradient conflicts and induced distribution shifts as potential causes of negative transfer and concluding that gradient conflicts are not necessarily tied to negative transfer.
2. The ForkMerge approach that the authors introduce is intuitive and simple, and the results are consistent across a variety of tasks.
3. Well-motivated and significant problem

**Weaknesses:**

1. In general, some critical details are omitted altogether in the main text, sometimes with no proper reference to the Appendix when details are missing. In particular:
- In Section 3, please spell out Painting and Quickdraw (and later, Real in Section 5) and describe DomainNet when first introducing the tasks. If it needs to be brief, the authors can at the very least include a pointer to Appendix C for dataset and task details.
- I also believe it is worth mentioning "We use ResNet-18 [8] pre-trained on ImageNet [3] for all experiments." in the main text rather than only noting it in the appendix.

2. I would have liked to see more overlap between works cited in the Experiments section and the works discussed in Related Work, especially discussion about and comparison of adaptive auxiliary- and multi-task learning strategies and settings. In particular, [36] and the original meta-learning paper [13]. A couple additional related works the authors may consider adding: https://arxiv.org/abs/2205.14082
 and https://arxiv.org/abs/2212.01378

**Questions:**

1. Can the authors describe in more detail the computational overhead of ForkMerge, especially wrt time and memory, and in relation to the existing approaches from Section 5? I am most curious about the comparison to meta-learning.

2. Do the authors predict that these results and the practicality of ForkMerge are limited to settings where one starts with a pretrained initialization?

3. re: definition 3.5, are there alternative measures of distribution shift that the authors have considered? I am personally somewhat skeptical of confidence scores as a basis for measuring distribution shift given that many models are poorly calibrated in the first place.

**Limitations:**

The authors test their method using only one model, ResNet-18 pre-trained on ImageNet, on image tasks. I won't push for additional experiments, but I would like to see the authors discuss what assumptions from this paper's experiments are expected to be critical for generalization of findings to other settings.

---

> ### Author Rebuttal · Authors · 2023-08-09
>
> We would like to sincerely thank Reviewer uS5H for providing insightful reviews and valuable comments. We have clarified the questions in the following response.
>
> **Q1:** Some critical details are omitted in the main text.
>
> Thank you for the feedback. Below, we outline our intended revisions:
>
> - We will make sure to spell out acronyms such as "Painting", "Quickdraw", and "Real" in the main text when first introducing them. Moreover, we will provide comprehensive descriptions of these tasks and the DomainNet dataset in the main text.
>
> - In our revised version, we will include a statement in the main text to mention that we utilize ResNet-18 pre-trained on ImageNet.
>
> **Q2**: More related work.
>
> We have carefully checked the recommended papers and agree that they are closely related to our research. In the camera-ready version, we will add the following discussion in the related work part.
>
> AANG [1] formulates a novel searching space of auxiliary tasks and adopts the meta-learning technique, which prioritizes target task generalization, to learn singlle-step task weightings.  This parallel finding highlights the importance of the target task generalization and we further introduce the multi-step task weightings to reduce the estimation uncertainty.
> Another parallel method, ColD Fusion [2], explores collaborative multitask learning and proposes to fuse each contributor's parameter to construct a shared model. In this paper, we further take into account the diversity of tasks and the intricacies of task relationships and derive a method for combining model parameters from the weights of task combinations.
>
> [1] *Dery, Lucio M., et al. AANG: Automating Auxiliary Learning. In ICLR, 2023.*
>
> [2] *Don-Yehiya, Shachar, et al. Cold fusion: Collaborative descent for distributed multitask finetuning. arXiv preprint.*
>
> **Q3**: Discussion on the computation cost.
>
> Please refer to $\text{question 2 (Q2)}$ of our global rebuttal.
>
> **Q4**: Is ForkMerge limited to settings where one starts with a pretrained initialization?
>
> The practicality of ForkMerge is not confined to settings with a pretrained initialization. In our experimental evaluation, we also employed the AliExpress, CIFAR-10, and SVHN datasets without leveraging any pretraining, as reported in $\text{Table 5}$ and $\text{Table 6}$, respectively.
>
> However, we did choose to use pretrained models in the case of DomainNet and NYUv2. This decision was rooted in two key factors. First, we aimed to ensure a fair comparison with prior works. Second, while not a universal requirement, the utilization of pretrained models is a prevalent setting. In numerous real-world applications, the absence of pretraining can significantly degrade performance.
>
> **Q5**: Are there alternative measures of distribution shift that the authors have considered?
>
> In the field of machine learning, various measures of distribution shift have been introduced, such as the Maximum Mean Discrepancy [3] and $\mathcal{H}\Delta\mathcal{H}$-Divergence [4], which have found applications in classic problems like domain adaptation. However, extending these measures to the context of auxiliary task learning presents certain challenges.
>
> The primary hurdle arises from the fact that Maximum Mean Discrepancy and $\mathcal{H}\Delta\mathcal{H}$-Divergence are originally defined over marginal distributions. This poses difficulties when dealing with auxiliary task learning scenarios where distinct tasks share a common feature space, yet exhibit variations in their output spaces. To address this limitation, we define the distribution shift in the output space, leading to the formulation of Confidence Score Discrepancy.
>
> While it is acknowledged that a deep learning model's predictive confidence on a specific data point might not always align with correctness, we posit that the expected confidence over a distribution captures certain characteristics of that distribution. While model calibration remains a concern, our definition places more emphasis on the relative magnitude of confidence. For instance, if the expected confidence for a test distribution $D_1$ is lower than that of another test distribution $D_2$, we interpret this as indicating a larger distance between the test distribution $D_1$ and the training distribution.
>
> [3] *Arthur Gretton, Karsten M. Borgwardt, Malte J. Rasch, Bernhard Sch olkopf, and Alexander Smola. A kernel two-sample test. In JMLR, 2012.*
>
> [4] *Ben-David, S., Blitzer, J., Crammer, K., and Pereira, F. Analysis of representations for domain adaptation. In NeurIPS, 2007.*
>
> **Q6:** Assumption of findings.
>
> - $\text{Finding 1 (Section 3.1)}$: We believe that the phenomenon we observed, where negative transfer is not solely attributed to gradient conflicts and vice versa, holds general relevance. This observation is supported by numerous instances, such as L2 regularization mentioned in the paper.
>
> - $\text{Finding 2 (Section 3.2)}$: The assertion that negative transfer is more likely when auxiliary tasks increase distribution shift between training and test data for the target task aligns with the principles of typical supervised learning. The validation of this assertion indeed presents certain challenges. To address this concern, we introduced the concept of Confidence Score Discrepancy, which quantifies the joint distribution distance in auxiliary task learning. It is important to note that the application of this concept assumes the presence of a confidence measure within the model. In cases such as regression tasks like the NYUv2 dataset, where this confidence measure is not as well-defined, its applicability may be limited. This is one of the reasons why we chose to conduct our analysis experiments on the DomainNet dataset, as it offers a suitable environment for investigating these intricacies.
>
> In summary, we hypothesize that while certain aspects may have limitations in diverse scenarios, the principles underlying them remain widely applicable.

---

> > ### Comment · Reviewer_uS5H · 2023-08-21
> >
> > Thank you for the detailed response and additional context! Updating my score to 7 assuming the authors will incorporate the discussions and clarifications into the camera ready if accepted.

---

> > > ### Author Response · Authors · 2023-08-22
> > > **Thanks for the Reviewer's Reply**
> > >
> > > We'd like to thank Reviewer uS5H again for providing an impressively insightful pre-rebuttal review, which has enabled us to make an effective response. We'd also thank you for carefully judging our feedback and acknowledging our work in the final review. **We will guarantee to incorporate the discussions and clarifications into the next version.**

---

### Official Review · Reviewer_QCm1 · 2023-07-26

**Soundness:** 2 fair
**Presentation:** 3 good
**Contribution:** 1 poor
**Rating:** 4
**Confidence:** 2

**Summary:**

Auxiliary-Task-Learning (ATL) has been studied from the perspective of optimization, which aims to improve the performance of the target task by leveraging similar tasks. However, ATL can sometimes suffer from negative transfer, where the performance of the target task actually decreases when auxiliary tasks are added. In this paper, the authors take a more holistic perspective by considering both optimization and generalization. They propose a new method called ForkMerge that is able to resolve negative transfer and improve the performance of the target task.

**Strengths:**

* ForkMerge is a simple and practical algorithm that demonstrates good empirical observation.
* Many analytic figures help readers to understand the negative transfer better although they are limited to DomainNet dataset.


**Weaknesses:**

* There seems to be some logical jump between the two observations listed in Section 3 and the proposed algorithm in Section 4. It is quite unclear how they are connected directly, and it’ll be great to have some theoretical formulation for the connection (at least in high-level). For example: why dynamically adjusting \lambda is superior to the previous method? How does this address the generalization problem? I think this limits the contribution of this work.
* ForkMerge essentially optimizes a fork of the given model and trains them separately multiple times; hence, it does require significantly larger compute during the model training.


**Questions:**

* As discussed in the weaknesses section, the connection between the two main observations to the ForkMerge is not strong. Can you provide a high level theoretical analysis of how ForkMerge can make better generalization not just based on the empirical analysis?
* Forking a large model is very expensive. Did authors consider a simple ensemble based approach which should be also similar in their compute (not the actual model weight merging rather than prediction level)?


**Limitations:**

I don’t see the negative societal impact of this work.

---

> ### Author Rebuttal · Authors · 2023-08-09
>
> We would like to sincerely thank Reviewer QCm1 for providing insightful reviews and valuable comments. We have clarified the questions in the following response.
>
> **Q1:** Logical Connection Enhancement.
>
> Please refer to $\text{question 1 (Q1)}$ of our global rebuttal.
>
> **Q2**: Why ForkMerge is superior to the previous method?
>
> **Enhanced Generalization.**
>
> While methods like GCS adjusted $\lambda$ based on the training data at each step, this primarily tackled optimization-level challenges without always guaranteeing improved generalization. The dynamic $\lambda$ adaptation in ForkMerge inherently captures the importance of generalization by directly considering the target validation performance during each merge step.
>
> **Mitigation of Estimation Uncertainty.**
>
> Approaches such as Auto-$\lambda$, which attempted to estimate $\lambda$ from a single gradient update, face challenges due to estimation fluctuations. The negative impact of inaccurately estimated $\lambda$ values on model parameters might be particularly noticeable. ForkMerge addresses this by using a longer time for $\lambda$ estimation, which reduces the negative impact of estimation errors on model parameters. The introduction of branch merging within our framework further strengthens the training process against inaccuracies in $\lambda$ estimation, making the algorithm more robust.
>
> **Efficient Computation.**
> The adoption of techniques such as Grid Search, which entailed exhaustive training and $\lambda$ tuning, led to escalating computational demands as the number of auxiliary tasks increased, resulting in exponential complexity. In contrast, ForkMerge dynamically adjusts $\lambda$ during each merging step, significantly reducing the complexity associated with exhaustive search. Furthermore, our approach mitigates the problem of suboptimal solutions that can arise due to fixed auxiliary task weights, as indicated in $\text{Figure 3 in Appendix D.1}$, and achieves a better trade-off between performance and computation cost.
>
> We hope these insights provide a clearer perspective on the advantages of ForkMerge's dynamic $\lambda$ adjustment mechanism and its superiority over previous methods.
>
> **Q3:** Concern on larger compute during the model training.
>
> Please refer to $\text{question 2 (Q2)}$ of our global rebuttal.
>
> **Q4**: Consider an ensemble-based approach focusing on merging the prediction?
>
> In this response, we'd like to highlight the distinct advantages of our ForkMerge algorithm in comparison to the suggested ensemble approach, while addressing the aspect of computational requirements.
>
> While ensemble methods indeed provide a means to combine multiple models, especially at the prediction level, it's essential to note that the computational costs associated with ensemble techniques during the testing phase can be substantial. Ensemble methods require making predictions using each individual model and then aggregating these predictions, leading to increased inference time and resource utilization.
>
> Thus, we have explored an alternative: conducting ensemble learning with multiple models during the training phase and distilling them into the target task. We use the term Pred-ensemble to represent this method. The experimental results on the DomainNet dataset are presented in $\text{Table 3 of our global rebuttal pdf}$. The Pred-ensemble method effectively improves the performance on $4$ of $6$ datasets, yet still lags behind our proposed ForkMerge on all tasks.

---

> > ### Comment · Reviewer_QCm1 · 2023-08-17
> >
> > Thanks authors for the detailed response. I have looked at the rebuttal (for my review and also the global), and the rebuttal does not fully address my concerns; hence I'll be keeping my current score.
> >
> > To be more specific, the logical connection still seems to be a bit hazy to me. I looked at Global Rebuttal #1, but I am not fully convinced. For example, "hyperparameter to be properly tuned -> let's dynamic change" is not well connected. Why is the model so sensitive with the hyperparameters? What model property is inducing this? Can we make modifications for the modeling assumptions so that this can be handled differently? These are not discussed well.

---

> > > ### Author Response · Authors · 2023-08-19
> > > **Replying to Reviewer**
> > >
> > > Thanks again for your dedication to reviewing our paper. We will provide additional clarification in this response.
> > >
> > > Firstly, it's important to clarify that **ForkMerge is inspired by, but not a direct extension of, the findings presented in $\text{Section 3}$.** ForkMerge is rooted in two fundamental principles of machine learning: the linear combination of losses in auxiliary task learning (as detailed in $\text{Equation 1}$) and the utilization of stochastic gradient descent for optimization (as outlined in $\text{Equation 5}$). Through these foundational principles, we naturally arrive at the concept of linearly combining model parameters (elaborated upon in $\text{Appendix A}$). Hence, the insights in $\text{Section 3}$ offer an intuitive explanation for the approach in $\text{Section 4}$, and even in the absence of $\text{Section 3}$, ForkMerge could still be derived through theoretical reasoning.
> > >
> > > **Q1:** Logical connection between our findings and proposed method.
> > >
> > > - $\text{Section 3.1}$ revisits the issue of gradient conflict and concludes that it's not necessarily correlated with negative transfer. **Thus, unlike prior methods, our approach does not initiate from the standpoint of gradient conflict.**
> > > - $\text{Section 3.2}$ underscores the importance of considering generalization.
> > >
> > > Drawing from these two points, ForkMerge leverages target task validation error for $\lambda$ selection and deduces the form of model parameter interpolation, as illustrated in $\text{Equation 7}$.
> > >
> > > Building upon these findings, our approach further:
> > >
> > > - Extends to dynamically adjusting $\lambda$ due to the evolving importance of different tasks during training (refer to $\text{Appendix D.1 Figure 3}$).
> > > - Employs longer time intervals for $\lambda$ estimation to reduce noise during the estimation process.

---

> > > ### Author Response · Authors · 2023-08-19
> > > **Replying to Reviewer**
> > >
> > > **Q2:** Why is the model so sensitive to the hyperparameters? What model property is inducing this?
> > >
> > > The sensitivity of the model to the weights assigned to different tasks is an inherent aspect of auxiliary-task learning. These task weights serve as crucial hyperparameters that directly influence the optimization objectives of the model.
> > >
> > > In the realm of auxiliary task learning and multi-task learning, numerous studies focus on tuning the weights of different tasks to attain optimal model performance. For instance:
> > >
> > > - **Uncertainty Weighting (UW)** [1] employs task uncertainties to weight the loss functions, effectively balancing the significance of various tasks.
> > > - **Dynamic Weighted Averaging (DWA)** [2] utilizes the decreased rate of task losses over time to weight the loss function dynamically.
> > > - **Gradient-Cosine Similarity (GCS)** [3] and **Auto-$\lambda$** [4] estimate dynamic task weights within a single iteration, using gradient cosine similarity and finite difference approximation, respectively.
> > >
> > > Furthermore, $\text{Section 3}$ of our paper provides a comprehensive exploration of the considerable impact that task weights have on the model from different perspectives:
> > >
> > > - **Multitask Optimization Perspective:** The task weights govern the optimization goal. Setting the auxiliary task weight to $0$ disregards the auxiliary task, while an infinitely large weight would cause the auxiliary task to dominate the primary task. This establishes the existence of an optimal equilibrium.
> > > - **Distribution Perspective**: The task weights influence the training distribution, as illustrated in $\text{Figure 3}$. A proper  selection of task weights can better fit the testing distribution of the target task.
> > >
> > > [1] *Alex Kendall, Yarin Gal, and Roberto Cipolla. Multi-task learning using uncertainty to weigh losses for scene geometry and semantics. In CVPR, 2018.*
> > >
> > > [2] *Shikun Liu, Edward Johns, and Andrew J Davison. End-to-end multi-task learning with attention. In CVPR, 2019.*
> > >
> > > [3] *Yunshu Du, Wojciech M Czarnecki, Siddhant M Jayakumar, Mehrdad Farajtabar, Razvan Pascanu, and Balaji Lakshminarayanan. Adapting auxiliary losses using gradient similarity. arXiv preprint arXiv:1812.02224, 2018.*
> > >
> > > [4] *Shikun Liu, Stephen James, Andrew J Davison, and Edward Johns. Auto-lambda: Disentangling dynamic task relationships. In TMLR, 2022.*
> > >
> > > **Q3:** Can we make modifications to the modeling assumptions so that this can be handled differently?
> > >
> > > The model's sensitivity to task weights stems from the fundamental intricacies of auxiliary-task learning, where careful consideration is required to strike a balance between mitigating overfitting and managing potential negative transfer.
> > >
> > > An intuitive solution could involve employing larger models to address this challenge, yet this approach brings forth its own set of complexities.
> > > Apart from the additional computational costs and memory overhead that come with larger models, a critical concern arises in the form of increased susceptibility to overfitting. Thus, a delicate trade-off between mitigating overfitting, a challenge ingrained in single-task learning, and the introduction of potential negative transfer due to auxiliary tasks must be meticulously weighed.
> > >
> > > Indeed, during the rebuttal phase, we delved into the prospect of employing larger models to examine this hypothesis. Specifically, we substituted the backbone network with ViT-Base [5], which has been pretrained on ImageNet 21K. We then conducted experiments on the DomainNet dataset, as outlined in $\text{Section 5.2}$. The results, as presented in $\text{Table 2 of our global rebuttal pdf}$, yield insightful observations:
> > >
> > > - ViT-Base demonstrates enhanced average accuracy through the equal weighting (EW) method, as compared to single-task learning. We posit that this improvement can be attributed to the data-hungry nature of vision transformers [5], wherein the advantages of auxiliary tasks in alleviating overfitting could potentially outweigh any interference introduced by the auxiliary tasks themselves.
> > > - ForkMerge consistently outperforms the comparison methods across all tasks, with an average accuracy of $73.3$ for ForkMerge, as opposed to $70.0$ for Post-train. This robust performance across various network architectures further substantiates the effectiveness of the ForkMerge approach.
> > >
> > > [5] *Dosovitskiy, Alexey et al. An Image is Worth 16x16 Words: Transformers for Image Recognition at Scale. In ICLR, 2021.*

---

### Official Review · Reviewer_G3Ms · 2023-07-26

**Soundness:** 3 good
**Presentation:** 3 good
**Contribution:** 2 fair
**Rating:** 6
**Confidence:** 3

**Summary:**

To fully leverage the knowledge from auxiliary tasks and mitigate negative transfer issues, this paper introduces ForkMerge, which automatically searches for varying task weights for auxiliary tasks by minimizing target validation errors. ForkMerge is evaluated under various settings, including multi-task learning, multi-domain learning, and semi-supervised learning. The results demonstrate that it outperforms existing methods and proves to be effective.

**Strengths:**

This paper demonstrates a well-organized structure. The authors have conducted experiments using diverse datasets and tasks, resulting in promising outcomes. Furthermore, the paper provides theoretical analysis of each component, offering valuable insights into their impact on performance. Overall, the clear structure and in-depth analyses make it an engaging and compelling read.

**Weaknesses:**

The paper lacks clear explanations for certain sentences, training details, and figure information. For example, there is a lack of clarity regarding the data division strategy for each branch and the effectiveness of the learned weights. Additional elaboration and detailed analysis in these areas would enhance the overall understanding and impact of the proposed approach.

**Questions:**

W1. The paper does not provide a clear explanation of how the method filters out harmful parameter updates to mitigate negative transfer after merging and synchronizing the parameters of each branch.

W2. Figure 3 would benefit from improved clarity, using a distinct color or different marks would be helpful. Additionally, why does the number of auxiliary task data points increase with the increasing $\lambda$?

W3. Lacks clarity regarding the data division strategy for each branch. It is not explicitly explained whether each branch is trained with a part of the data or full data. Moreover, the impact of increasing the number of branches on computational cost and its trade-off with efficiency and accuracy is not thoroughly addressed in the paper.

W4. I am curious about the comparison between the learned weights by ForkMerge and the optimal weights. For instance, in Figure 3 (b), for WNT tasks, do the learned weights resemble the optimal weights shown in the figure?

W5. It would be valuable to investigate and illustrate the trajectory of the learned weights over the course of training to understand their dynamics and convergence patterns. To gain a better understanding of the effectiveness of the ForkMerge approach, it would be insightful to compare the performance of fixed learned weights by ForkMerge with dynamically learned weights during training.



**Limitations:**

Limitations could be more clarified in the paper.

---

> ### Author Rebuttal · Authors · 2023-08-09
>
> We would like to sincerely thank Reviewer G3Ms for providing insightful reviews and valuable comments. We have clarified the questions in the following response.
>
> **Q1:** Clarification on the data division strategy for each branch.
>
> Depending on the characteristics of auxiliary task learning scenario, there are two circumstances.
>
> - In the case of the NYUv2 dataset, multiple tasks share the same input, but their outputs are different. In this setup, each branch in the ForkMerge algorithm has the same input data, which includes the entire dataset. The distinction between different branches solely lies in the task weighting.
> - In contrast to the NYUv2 scenario, for datasets like DomainNet, different tasks have both different inputs and outputs. In these cases, for each branch, if the task weighting of a specific task is set to $0$, the data from that particular task will not be used for training the corresponding branch.
>
> We appreciate the reviewer's feedback and will add clarification on the data division strategy for each branch in our camera-ready version.
>
> **Q2:** The trajectory of the learned weights over the course of training.
>
> We have visualized the trajectory of the learned weights in $\text{Figure 3 of Appendix D.1}$, which indicates the relative ratio of each forking branch is dynamic and varies from task to task.
>
> **Q3:** Clarification on the effectiveness of the learned weights. Comparison between the learned weights by ForkMerge and the optimal weights.
>
> **Effectiveness of the Learned Weights**.
>
> We would like to clarify that the learned weights of ForkMerge are a sequence of task weightings rather than a fixed one.
>
> - At each merging step, ForkMerge will search for an optimal task weighting based on the current learning status of all branches.
> - $\text{Figure 3 in Appendix D.1}$ visually demonstrates the dynamic behavior of task weighting during the training process in ForkMerge. As shown in the figure, there is no evidence to suggest that the task weighting will converge to a fixed value during training. Instead, the algorithm continuously adjusts the task weighting.
>
> Regarding the effectiveness of the learned sequence of task weighting, our experiments in $\text{Section 5}$ demonstrate that ForkMerge consistently outperforms existing methods on various benchmarks.
>
> **Comparison with Optimal Weights.**
>
> We are not pretty sure about the "optimal weights" mentioned in your question and we understand the possible reference to static weights obtained from grid search.
>
> - We have provided a comparison between ForkMerge and grid searching $\lambda$ on the NYUv2 dataset, as presented in $\text{Appendix D.4}$. In $\text{Figure 6 of Appendix D.4}$, we visualize the performance comparison of all methods with grid search. For grid search, there are a total of $27$ weighting configurations. It can be observed that ForkMerge achieves substantial improvement over grid search on all tasks.
>
> If there is any confusion or misunderstanding regarding the "optimal weights" mentioned in your question, we would be glad to answer any further questions and provide additional clarification.
>
> **Q4:** Comparison with fixed learned weights by ForkMerge.
>
> As clarified above, defining fixed learned weights for ForkMerge might not be straightforward and may not hold a meaningful interpretation. It is important to note that ForkMerge naturally generates a dynamic sequence of task weightings, continuously adapting during the training process. And our experiments in $\text{Appendix D.4}$ provide evidence that this dynamic task weighting mechanism outperforms grid searching static weights.
>
> **Q5:** Clarification on how the method filters out harmful parameter updates to mitigate negative transfer.
>
> As outlined in $\text{Section 4}$ of our paper, ForkMerge operates through an iterative process involving fork and merge steps:
>
> - During the fork step, each branch in ForkMerge is independently trained. It is in this step that harmful parameter updates might occur, potentially compromising the performance on the target task.
> - In the merge step, ForkMerge searches for the optimal task weighting combination of different branches. This mechanism empowers ForkMerge to dynamically adjust the task weighting for each branch based on its contribution to the overall performance. As a result, ForkMerge can decrease the weighting of branches where negative transfer occurs. In extreme cases, it can set the weighting to $0$, ignoring harmful parameters. After merging, the newly obtained parameters are synchronized across all branches.
>
> By synchronizing the new parameters to all branches, ForkMerge ensures that the harmful parameter updates experienced during the fork step are effectively filtered out. The filtering process occurs post-merging, allowing each branch to benefit from the collective knowledge while mitigating the influence of negative transfer.
>
> **Q6:** Why does the number of auxiliary task data points increase with the increasing $\lambda$ in Figure 3?
>
> As discussed in $\text{Section 3.2}$, adjusting $\lambda$ will change the data distribution that the model is fitting. When the weighing hyper parameter of the auxiliary task increases, the effect of the auxiliary task on the interpolated distribution will also increase. As a result, to visualize the impact of $\lambda$ on the interpolated training distribution, we let the frequency of auxiliary task points be proportional to $\lambda$ (introduced in $\text{Appendix B.2}$).
>
> **Q7:** The impact of increasing the number of branches on computational cost and its trade-off with efficiency and accuracy is not thoroughly addressed in the paper.
>
> Please refer to $\text{question 2 (Q2)}$ of our global rebuttal.

---

### Official Review · Reviewer_yZ6C · 2023-07-27

**Soundness:** 2 fair
**Presentation:** 3 good
**Contribution:** 2 fair
**Rating:** 6
**Confidence:** 4

**Summary:**

This paper strives to mitigate negative transfer in auxiliary-task learning by optimizing the coefficients assigned to auxiliary tasks. By conducting an empirical investigation into the factors contributing to negative transfer, this paper reveals two interesting findings. Based on the findings, a new approach named ForkMerge is proposed to mitigate negative transfer and boost the performance of auxiliary-task learning. Extensive experiments demonstrate the effectiveness of the proposed approach.

**Strengths:**

+The problem is well-defined and well-motivated.
+The findings seem interesting
+The proposed approach is reasonable and well presented.

**Weaknesses:**

-It is not clear how the new findings motivate the proposed ForkMerge.

-I'm curious about the performance when directly trying various lambdas (grid search) during multiple training sessions. Although the proposed method is more efficient, there is a concern about potential accuracy trade-offs. I seek to understand if the proposed approach sacrifices accuracy compared to the direct lambda variation method.

-The simple post-train method leads to superior performance over other complex approaches, which makes me doubt the significance of research efforts in this area over the past years. This paper claims that the main drawback of post-train method is that it fails to consider the task relationship in the pre-training phase, and suffers from forgetting during fine-tuning. I’m wondering the performance of the post-train method if considering the task relationship in the pre-training phase and preventing forgetting during fine-tuning such as by distillation. In this scenario, if the post-train method manages to surpass the proposed ForMerge approach, considering ForMerge’s slight improvement in most tasks compared to Post-train.

-The experimental evaluation is limited to small datasets, such as NYUv2 and CIFAR-10, and employs a relatively small network like ResNet-50 (DeepLabV3+). As a result, there is uncertainty regarding the efficacy of the proposed method on larger datasets and powerful networks like transformers, which already excel in single-target tasks. Consequently, assessing the true significance of the proposed method in the current era of deep learning becomes challenging.

-Too may abbreviations make paper hard to follow in some paragraphs.

**Questions:**

See Weaknesses

**Limitations:**

Not sure

---

> ### Author Rebuttal · Authors · 2023-08-09
>
> We would like to sincerely thank Reviewer yZ6C for providing insightful reviews and valuable comments. We have clarified the questions in the following response.
>
> **Q1:** It is not clear how the new findings motivate the proposed ForkMerge.
>
> Please refer to $\text{question 1 (Q1)}$ of our global rebuttal.
>
> **Q2:** Comparison with directly grid searching task weighting $\lambda$.
>
> In $\text{Figure 6 of Appendix D.4}$, we present the comparison of all methods with grid search. For grid search, there are a total of $27$ weighting configurations. Our observations are as follows:
>
> - Existing methods typically yield performance trade-off points that lie along the scalarization Pareto front.
> - ForkMerge produces results that are distant from the Pareto front, resulting in substantial performance gains over the grid search technique.
> - This improvement can be attributed to the fact that grid search tends to converge towards suboptimal solutions due to its reliance on fixed auxiliary task weights. Conversely, ForkMerge possesses the capability to dynamically adjust task weights continuously, as depicted in $\text{Figure 3 of Appendix D.1}$.
>
> **Q3:** Comparison with improved Post-train method.
>
> To address the reviewer's concern, we will first discuss the Post-train method and then present additional experimental results with an improved version of Post-train.
>
> **Discussion on Post-train Method.**
>
> - Post-train can be seen as a specific instance within ForkMerge's framework. In the first half of the training period, only branches with equal task weighting are employed, while in the second half of the period, only branches with single task weighting are used.
> - Our experimental results demonstrate the superiority of ForkMerge over Post-train across all benchmark tasks. Specifically, ForkMerge achieves substantial performance gains in CTR and CTCVR prediction tasks ($\text{Table 5}$: $+1.30\\%$ *v.s.* $+0.14\\%$) and the semi-supervised learning task ($\text{Table 6}$: $+46.3\\%$ *v.s.* $+30.4\\%$).
> - Lastly, when the performance of single task learning (STL) is worse than equal weighting (EW), the fixed post-training strategy may not yield performance gains. For example, when employing ViT-Base as the backbone network on the DomainNet dataset, where ViT is particularly data-hungry, the performance of the Post-train approach falls behind that of EW and ForkMerge. For comprehensive results, please consult $\text{Table 2 of our global rebuttal pdf}$.
>
> **Additional Experiments.**
>
> We enhance the Post-train method by adopting the Knowledge Distillation (KD) technique [1] to preserve the knowledge. $\text{Table 1 of our global rebuttal pdf}$ presents the results on the DomainNet dataset. As evident from the table, ForkMerge outperforms the improved Post-train method.
>
> [1] *Hinton, Geoffrey, Oriol Vinyals, and Jeff Dean. Distilling the knowledge in a neural network. In NeurIPS Workshop, 2014.*
>
> **Q4:** Limitation on the scale of evaluation datasets.
>
> - We adopt the NYUv2 dataset in scene understanding task, CIFAR10, and SVHN datasets in semi-supervised learning task as they are widely used in the auxiliary task learning literature. By doing so, we can provide a more fair and meaningful comparison with prior methods.
> - Additionally, as shown in $\text{Section 5}$, we further conduct experiments on medium scale DomainNet dataset (about $0.6$M images) and large scale AliExpress dataset (over $100$M records). ForkMerge clearly outperforms existing methods on both datasets, affirming its efficacy across datasets of different magnitudes.
>
> **Q5:** Experiments with advanced architectures such as transformers.
>
> To address the reviewer's concern, we replace the backbone network with ViT-Base [2] pretrained on ImageNet 21K and repeat the experiments on DomainNet of $\text{Section 5.2}$. $\text{Table 2 of our global rebuttal pdf}$ presents the results and we have the following observations.
>
> - With ViT-Base, the equal weighting (EW) method achieves higher average accuracy over single task learning. We conjecture that this improvement can be attributed to the data-hungry nature of vision transformers [2]. In this case, the benefits of reducing overfitting through the use of auxiliary tasks may outweigh the potential task interference problem.
> - ForkMerge outperforms compared methods on all tasks (average accuracy: ForkMerge $73.3\\%$ *v.s.* Auto-$\lambda$ $71.5\\%$), validating its efficacy across different network architectures.
>
> [2] *Dosovitskiy, Alexey et al. An Image is Worth 16x16 Words: Transformers for Image Recognition at Scale. In ICLR, 2021.*
>
> **Q6:** Too many abbreviations make the paper hard to follow in some paragraphs.
>
> Many thanks to the reviewer's feedback. We will update our camera-ready version as follows:
>
> - We will reduce the use of abbreviations.
>   - For instance, in the analysis part ($\text{Section 3}$), we will directly use the full concepts such as "Transfer Gain," "Weak Negative Transfer," "Strong Negative Transfer," and "Confidence Score Discrepancy" instead of their abbreviations.
>   - Besides, we will include a table that explains the meaning of all abbreviations used in the paper.
> - We will provide more details in the main text rather than solely discussing these issues in the Appendix.
>   - For instance, in the analysis part ($\text{Section 3}$), we will include a description of the DomainNet dataset, outline our dataset processing steps, and provide details about the network.
>   - In the method part ($\text{Section 4}$), we will elaborate on the data division strategy for each branch, provide details about how we prune branches and efficiently merge different branches to reduce computation costs. Additionally, we will provide a discussion comparing $\text{Equation 12}$ and $\text{Equation 13}$ to clarify their difference.
>   - In the experiment part ($\text{Section 5}$), we will present more training details, including the specific network architecture and the important hyperparameters.

---

> > ### Comment · Reviewer_yZ6C · 2023-08-22
> >
> > Thank the authors for their comprehensive response, which has effectively solved some of my major concerns. However, I do have an additional query regarding the performance of the post-train method.
> >
> > In Table 2 of the primary manuscript, the post-train method exhibits superior performance compared to EW and Auto_lambda. Nevertheless, when employing the more robust ViT-Base model, an intriguing shift occurs as shown in Table 2 of the global rebuttal. The post-train method's performance deteriorates, whereas the performance of the other two methods experiences a significant improvement. Could the authors kindly provide further elucidation regarding these observed outcomes?

---

> > > ### Author Response · Authors · 2023-08-22
> > > **Replying to Reviewer**
> > >
> > > We sincerely appreciate your feedback and are pleased to see that our response has addressed many of your concerns. We are grateful for the opportunity to provide further clarification on the additional question regarding the performance of the post-train method.
> > >
> > > In $\text{Section 5.2}$, our experiment results on DomainNet show that the Post-train method has consistently exhibited a slight advantage over STL. This trend can be attributed to several factors that contribute to the generalization ability of Post-train:
> > >
> > > - **Influence of Single-Task Fine-Tuning**: In datasets with a substantial volume of data, such as DomainNet, the final performance of a model is **notably influenced by the last stage of training**, which involves single-task fine-tuning. Consequently, when STL performs well, the Post-train method also benefits from single-task fine-tuning, resulting in better performance.
> > >
> > > - **Implicit Regularization from Pre-training**: The initial pre-training phase serves as a form of parameter initialization, offering implicit regularization that aids in optimizing the model. This initialization effect contributes to the enhanced generalization capability of the Post-train method compared to STL.
> > >
> > > However, we acknowledge the intriguing observation you have made when applying the more robust ViT-Base model, as highlighted in $\text{Table 2 of global rebuttal pdf}$. In this case, there is a shift in performance dynamics, where the Post-train method's performance deteriorates while both the EW and Auto-$\lambda$ methods experience significant improvements. This phenomenon can be attributed to the interaction between auxiliary tasks and the target task, which has varying implications across different model architectures:
> > >
> > > **Impact of Auxiliary Tasks**: The introduction of auxiliary tasks brings about a dual effect on the model's performance. On one hand, the additional supervision signals contribute to a reduction in the risk of overfitting, enhancing generalization. On the other hand, the joint distribution shift between the auxiliary tasks and the target task can lead to negative transfer, impacting performance adversely. In different scenarios, both of these effects are present, but one effect might be more pronounced.
> > >
> > > **Impact of Model Capacity:** The influence of the dual effects of auxiliary tasks is dependent on the specific scenario and model capacity.
> > >
> > > - For instance, with limited model capacity, as is the case with the ResNet101 architecture on DomainNet, the influence of task conflicts induced by auxiliary tasks becomes more pronounced. This results in the performance of the EW method being inferior to both STL and subsequently Post-train.
> > >
> > > - Conversely, when employing the Vision Transformer model, which boasts increased capacity, the risk of overfitting with limited data becomes more pronounced. This makes STL less effective and consequently leads to the EW method outperforming STL, causing the Post-train method to fall short of EW and Auto-$\lambda$.
> > >
> > > Across different scenarios, there exists an equilibrium point between EW and STL, driven by the dynamic interplay between the positive and negative effects of auxiliary tasks. **This equilibrium point represents the central aim of our ForkMerge algorithm: to uncover the optimal balance between these effects.**
> > >
> > > We believe that the consistent patterns observed across various scenarios underscore the utility and potential of the ForkMerge algorithm in identifying this equilibrium and leveraging the strengths of both STL and EW methods. We hope that this additional explanation provides a clearer understanding of the observed outcomes, and we remain open to any further inquiries you may have.
> > >
> > > Thank you once again for your valuable insights and the opportunity to enhance the clarity of our work.

---

### Author Rebuttal · Authors · 2023-08-09

We would like to sincerely thank all the reviewers for providing insightful reviews and valuable comments. Your reviews are of great importance to us in improving the quality of this work.

**In this global rebuttal, we aim to clarify the common questions from reviewers, and we have responded to each reviewer with a separate response for other questions. The full results of additional experiments are attached in the one-page pdf.**

**Q1:** How do the new findings motivate the proposed ForkMerge?

Below we outline the motivations behind the design of ForkMerge in light of our new findings.

- **[Finding]** $\text{Section 3.1}$ reveals that the presence of gradient conflict does not necessarily lead to negative transfer, as long as the hyperparameter $\lambda$ is appropriately tuned. Additionally, $\text{Section 3.2}$ emphasizes the importance of considering generalization to mitigate negative transfer effectively. **[Algorithm Design]** Based on these findings, we opt to dynamically adjust the hyperparameter $\lambda$ according to the target validation performance in ForkMerge ($\text{Section 4.1}$).
- **[Finding]** $\text{Section 3}$ highlights that in scenarios where weak negative transfer (WNT) occurs, selecting an appropriate value for $\lambda$ can alleviate the problem. However, in cases of strong negative transfer (SNT), setting $\lambda$ to $0$ becomes necessary. **[Algorithm Design]** In each merging step of ForkMerge, we perform a search step to identify the optimal value of $\lambda$, which effectively mitigates weak negative transfer. Furthermore, for instances of strong negative transfer, ForkMerge is able to set $\lambda$ to 0 to prevent negative transfer ($\text{Section 4.1}$). Additionally, we have introduced a pruning mechanism to remove SNT forking branches, thus reducing the computation cost ($\text{Section 4.2}$).
- **[Finding]** $\text{Section 3.2}$ indicates that negative transfer is likely to occur when the introduced auxiliary task enlarges the distribution shift between the training and test data for the target task. To address this issue, it is crucial to select auxiliary tasks that decrease the distribution shift between training and test data for the target task. **[Algorithm Design]** To address the distribution shift problem, the general form of ForkMerge constructs mixture distributions that comprise diverse data shifts relative to the target distribution. Subsequently, models trained on these different distributions are combined dynamically to approach the optimal parameters ($\text{Section 4.2}$).

**Q2:** Concern about the computation cost and trade-off between efficiency and accuracy.

**Clarification on the Computation Cost.**

Firstly, we have developed several techniques to reduce computation cost. Below, we provide a detailed explanation:

- **Pruning Strategy:** As introduced in $\text{Section 4.2}$, we can prune the forking branches with $\Lambda_k=0$ and only keep the branches with the largest $K'<K$ values in $\Lambda$ after the early merge step, where $K$ represents the total number of tasks.
  - To illustrate the effectiveness of the pruning strategy, we present results in $\text{Section 5.2}$ on Auxiliary-Domain Image Recognition and CTR and CTCVR Prediction tasks. For instance, on the CTR and CTCVR Prediction task, we initially construct up to $8$ branches with different task weights, but after the first merge step, we prune them to $3$ branches, achieving considerable computational savings.
- **Greedy Strategy in the Merge Step:** In $\text{Algorithm 2 of Appendix A.2}$, we introduce a greedy strategy during the merge step. This modification reduces the computation complexity from exponential to linear complexity when searching for optimal task weighting.
- **Validation Set Sampling:** As mentioned in $\text{Appendix A.1}$, the costs associated with estimating validation performance $\hat{\mathcal{P}}$ in the search step are usually negligible. However, when the validation set size is relatively large, we can resort to sampling to reduce the computational cost further.

Further, we have conducted analysis of the computation cost in $\text{Appendix D.2}$:

- Although only one model is optimized in most previous auxiliary-task learning methods, their computational costs are not necessarily $\mathcal{O}(1)$. For example, gradient balancing methods require computing gradients of each task, thus leading to $\mathcal{O}(K)$ complexity. In addition, calculating the inner product or norm of the gradients will bring a calculation cost proportional to the number of network parameters.
- To support our claims, we access the actual training time across methods on NYUv2. As depicted in $\text{Figure 4 of Appendix D.2}$, ForkMerge does not require more training time than other auxiliary task learning methods, including GCS, OL_AUX, ARML, and Auto-$\lambda$.

**Memory Utilization.**
In terms of memory usage, it's important to note that the optimization of each branch within ForkMerge is entirely independent. This enables us to load only the model parameters corresponding to the particular branch being trained at any given time. Consequently, the storage requirements are comparable to those of single task learning (STL), resulting in minimal memory overhead.

**Trade off between Efficiency and Accuracy.**

In $\text{Section 5.2}$, we provide an analysis on DomainNet to explore the trade-off between efficiency and accuracy. Results in $\text{Table 4}$ reveal that as the number of branches increases, the gain by auxiliary tasks will enlarge while the gain brought by each branch will reduce.

In this paper, we propose to address this trade-off through the pruning strategy, which allows users to customize the number of branches based on their specific needs and available computational resources.

---

### Decision · Program_Chairs · 2023-09-21

**Decision:**

Accept (poster)

**Comment:**

The paper presents a way to improve target task learning via auxiliary tasks. In particular, the “ForkMerge” procedure trains two parallel branches, one on target task and another with target + auxiliary task. The weights of the two branches are linearly interpolated based on validation performance. Experiments show the efficacy of the proposed method over several settings.

The paper is predominantly empirical and the reviewers were initially split in their opinions. There were concerns about the validity of the results with respect to larger datasets/architectures, sensitivity to hyper-parameters, and motivation of the method. Most concerns are sufficiently addressed during the discussion. Most reviewers recommended acceptance of the paper.